# QLCODER: A QUERY SYNTHESIZER FOR STATIC ANALYSIS OF SECURITY VULNERABILITIES

**Claire Wang**
University of Pennsylvania
cdwang@seas.upenn.edu

**Ziyang Li**
John Hopkins University
ziyang@cs.jhu.edu

**Saikat Dutta**
Cornell University
saikatd@cornell.edu

**Mayur Naik**
University of Pennsylvania
mhnaik@upenn.edu

## ABSTRACT

Static analysis tools provide a powerful means to detect security vulnerabilities by specifying *queries* that encode vulnerable code patterns. However, writing such queries is challenging and requires diverse expertise in security and program analysis. To address this challenge, we present *QLCoder* – an agentic framework that automatically synthesizes queries in CodeQL, a powerful static analysis engine, directly from a given CVE metadata. QLCoder embeds an LLM in a synthesis loop with execution feedback, while constraining its reasoning using a custom MCP interface that allows structured interaction with a Language Server Protocol (for syntax guidance) and a RAG database (for semantic retrieval of queries and documentation). This approach allows QLCoder to generate syntactically and semantically valid security queries. We evaluate QLCoder on 176 existing CVEs across 111 Java projects. Building upon the Claude Code agent framework, QLCoder synthesizes correct queries that detect the CVE in the vulnerable but not in the patched versions for 53.4% of CVEs. In comparison, using only Claude Code synthesizes 10% correct queries. Our generated queries achieve an F1 score of 0.7. In comparison, the general query suites in IRIS (a recent LLM-assisted static analyzer) and CodeQL only achieve F1 scores of 0.048 and 0.073, highlighting the benefit of QLCoder's specialized synthesized queries. QLCoder is available at https://github.com/neuralprogram/qlcoder.

## 1 INTRODUCTION

Security vulnerabilities continue to grow at an unprecedented rate, with over 40,000 Common Vulnerabilities and Exposures (CVEs) reported in 2024, 28,961 CVEs reported in 2023, and 25,059 CVEs reported in 2022 (CVE, 2025). Static analysis, a technique to analyze programs without executing them, is a common way of detecting vulnerabilities. Static analysis tools such as CodeQL (GitHub, 2025b), Semgrep (Semgrep, 2023), and Infer (Meta, 2025) are widely used in industry. They provide domain-specific languages that allow specifying vulnerability patterns as queries. Such queries can be executed over structured representations of code, such as abstract syntax trees, to detect potential security vulnerabilities.

Despite their widespread use, existing query suites of static analysis tools are severely limited in coverage of vulnerabilities and precision. Extending them is difficult even for experts, as it requires knowledge of unfamiliar query languages, program analysis concepts, and security expertise. Incorrect queries can produce false alarms or miss bugs, limiting the effectiveness of static analysis. Correct queries can enable reliable detection of real vulnerabilities, supporting diverse use-cases such as regression testing, variant analysis, and patch validation, among others (Figure 1).

Meanwhile, CVE databases (MITRE, 2025; NIST, 2025; GitHub, 2025d) provide rich information about security vulnerabilities, including natural language descriptions of vulnerability patterns and records of buggy and patched versions of the affected software repositories. This resource remains largely untapped in the automated construction of static analysis queries. Recent advances in LLMs, particularly in code understanding and generation, open up the possibility of leveraging this informa-

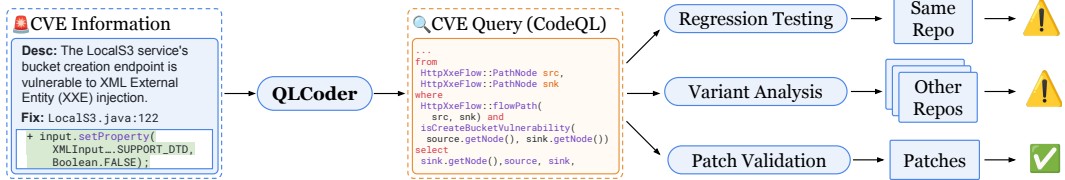

Figure 1: A CodeQL query capturing a vulnerability pattern is synthesized by QLCoder from an existing CVE and subsequently reused for regression testing, variant analysis, or patch validation.

tion to automatically synthesize queries from CVE descriptions, thereby bridging the gap between vulnerability reports and practical detection tools.

Synthesizing such queries poses significant challenges. The syntax of static analysis query languages is low-resource, richly expressive, and evolves continually. A typical query, such as the one in Figure 2(b) specifying a global dataflow pattern leaves ample room for errors in describing predicates for sources, sinks, sanitizers, and taint propagation steps. Even if the generated syntax is correct, success is measured by whether the query can identify at least one execution path traversing the bug location in the vulnerable version while producing no matches in the patched version. Achieving this requires understanding the CVE context at the level of abstract syntax trees, such as code differences that introduce a sanitizer to prevent a flow from a source to a sink. Complicating matters further, reasoning about the code changes alone is often insufficient: sources, sinks, and taint propagation steps may reside in parts of the codebase far from the modified functions or files, and the vulnerability itself may involve non-trivial dataflow chains across these components. Thus, a correct query must not only integrate information from multiple locations across the program but also capture the intricate propagation patterns to accurately characterize the vulnerability.

In this paper, we present **QLCoder**–an agentic framework that synthesizes queries in CodeQL, a powerful static analysis engine, directly from a given CVE metadata. We select CodeQL because it has the richest query language, which allows capturing complex inter-procedural vulnerability patterns. QLCoder addresses the above challenges by embedding an LLM in a structured synthesis loop that incorporates execution feedback to verify query correctness and allows interactive reasoning using a custom MCP (Model Context Protocol) interface. The MCP interface constrains the model's reasoning using a Language Server Protocol (for syntax guidance) and a vector database of CodeQL queries and documentation (for semantic guidance). By combining these capabilities, QLCoder avoids common pitfalls of naive LLM-based approaches, such as producing ill-formed queries, hallucinating deprecated constructs, or missing subtle vulnerability patterns, and instead produces queries that are both syntactically correct and semantically precise.

We evaluate QLCoder on CWE-Bench-Java (Li et al., 2025b), which comprises 176 CVEs across 111 Java projects. These CVEs span 42 different Common Weakness Enumeration (CWE) categories and the projects range in size from 0.01 to 1.5 MLOC. To account for model training cut-offs, we include 65 CVEs reported during 2025 and target a recent CodeQL version 2.22.2 (July 2025). Using the Claude Code agent framework, QLCoder achieves query compilation and success rates of 100% and 53.4%, compared to 19% and 0% for our best agentic baseline, Gemini CLI. Further, our generated queries have an F1 score of 0.7 for detecting true positive vulnerabilities, compared to 0.048 for IRIS (Li et al., 2025b), a recent LLM-assisted static analyzer, and 0.073 for CodeQL.

We summarize our main contributions:

- **Agentic Framework for CVE-to-Query Synthesis.** We present QLCoder, an agentic framework that translates CVE descriptions into executable CodeQL queries, bridging the gap between vulnerability reports and static analysis. QLCoder introduces a novel integration of execution-guided synthesis, semantic retrieval, and structured reasoning for vulnerability query generation.
- **Evaluation on Real-World Repositories and CVEs.** We evaluate QLCoder on 176 CVEs in Java projects, covering 42 vulnerability types (CWEs) from CWE-Bench-Java. Each project involves complex inter-procedural vulnerabilities spanning multiple files. We show how QLCoder can successfully identify sources, sinks, sanitizers, and taint propagation steps, and refine queries to ensure they raise alarms on vulnerable versions while remaining silent on patched versions.
- **Comparison with Baselines.** We compare QLCoder against state-of-the-art agent frameworks and show that QLCoder achieves substantially higher compilation, success, and F1 scores. We

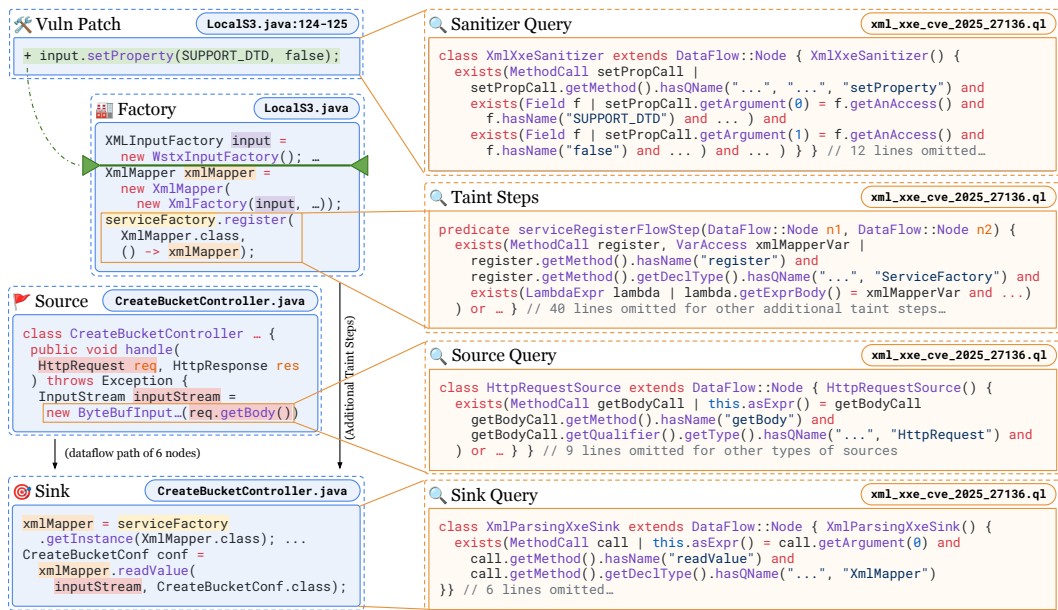

(a) The vulnerable dataflow snippets and the patch, which adds configuration to XMLInputFactory.

(b) Snippets of the synthesized vulnerability query by QLCoder capturing the patterns of dataflow source, sink, taint steps, and the sanitizer indicated by the vulnerability patch.

```
module HttpXxeFlow = TaintTracking::Global<HttpXxeFlowConfig>; // Config: HttpRequestSource, XmlParsingXxeSink, …
import HttpXxeFlow::PathGraph

from HttpXxeFlow::PathNode source, HttpXxeFlow::PathNode sink
where HttpXxeFlow::flowPath(source, sink)
select sink.getNode(), source, sink, "HTTP request data flows to XML parser…", source.getNode(), "HTTP request entry point"
```

(c) The synthesized CodeQL path query that ties everything together.

Figure 2: Illustration of vulnerability CVE-2025-27136 in repository Robothy/local-s3 which exhibits an XML External Entity Injection weakness (CWE-611). When the XmlMapper is not configured to disable *Document Type Definition* (DTD), the function readValue may declare additional entities, allowing hackers to inject malicious behavior.

also compare QLCoder's synthesized queries with state-of-the-art static analysis frameworks and show that our queries are more precise and have higher recall.

## 2 ILLUSTRATIVE EXAMPLE

We illustrate the challenges of vulnerability query synthesis using CVE-2025-27136, an XML External Entity Injection (XXE) bug found in the repository Robothy/local-s3. Figure 2 depicts the vulnerability snippets, the patch, and the synthesized CodeQL query generated by QLCoder.

**Vulnerability context.** The vulnerability arises when the XmlMapper object is used to parse user-provided XML data (Figure 2a). In the vulnerable code, XmlMapper.readValue is called on the HTTP request body without disabling support for *Document Type Definitions* (DTDs). As a result, an attacker can inject malicious external entity declarations into the input stream, enabling server-side request forgery (SSRF) attacks, allowing for access to resources that should not be accessible from external networks, effectively leaking sensitive information. The patch mitigates the issue by configuring the underlying XMLInputFactory with the property SUPPORT_DTD=false.

**Synthesizing the query.** The CodeQL query that can effectively capture the vulnerability pattern needs to incorporate 1) sources such as HttpRequest.getBody calls where untrusted malicious information enters the program, 2) sinks such as invocations of XmlMapper.readValue, where the XXE vulnerability is manifested, 3) additional taint steps related to how the XmlMapper is

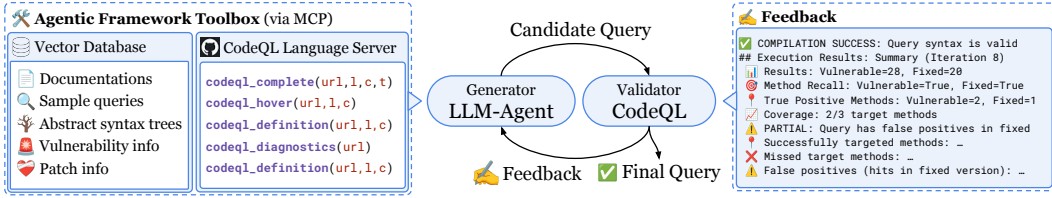

Figure 3: Overall pipeline of QLCoder's iterative synthesis loop between an agentic query generator and a CodeQL-based validator. The generator uses a vector database and our CodeQL Language Server as tools while the validator produces compilation, execution, and coverage feedback.

constructed and configured, involving non-trivial interprocedural flows spanning multiple files, and 4) sanitizers such as calls to `setProperty(SUPPORT_DTD, false)`, so that we know that no alarm should be reported after the vulnerability has been fixed.

In general, the synthesized query must connect all these components to be able to detect the bug in the vulnerable program, while not reporting the same alarm after the vulnerability has been fixed. Figure 2b shows all the components of the CodeQL query (simplified), capturing their individual syntactic patterns. Lastly, Figure 2c connects all these components into a coherent path query by using CodeQL's `TaintTracking::Global<.>::PathGraph` and the SQL-like `from-where-select` query, which returns the exact path from source to sink.

**Challenges and solutions.** Vulnerability query synthesis must overcome several tightly-coupled challenges. We hereby state the challenges and explain how QLCoder addresses them.

- *Rich expressiveness and fragility of syntactic patterns.* CodeQL is powerful but syntactically intricate: small mistakes in predicate names, qualifiers, or AST navigation often produce syntactically valid yet semantically useless queries. QLCoder mitigates this fragility through its Language Server Protocol (LSP) interface for syntax guidance and RAG database for semantic retrieval of existing CodeQL queries and documentation. These structured interactions guide predicate selection and AST navigation during synthesis, reducing off-by-name and version-mismatch errors.

- *Inter-procedural taint propagation across a large codebase.* Sources, sinks, and sanitizers typically live in different modules or files and are connected by nontrivial inter-procedural flows (lambdas, factory patterns, etc.). While CodeQL provides robust inter-procedural analysis for many common patterns, gaps in dataflow still require bridging via additional taint propagation steps. Through its custom MCP interface, QLCoder performs structured reasoning to discover candidate program points, synthesize custom taint-step predicates (e.g., service registration), and compose them into a CodeQL path query that tracks data across file and component boundaries.

- *Semantic precision: alarm on the vulnerable version, silence on the patched version.* A useful vulnerability query must not only parse correctly but also be discriminative. QLCoder enforces this semantic requirement directly during synthesis. Via an iterative refinement loop, the successful criteria states that in the fixed program, there should be no alarm being raised about the vulnerability. This incentivizes the agent to synthesize sanitizer predicates (e.g., the `setProperty` call) and use them to constrain the path query so that sanitizer presence suppresses the alarm. The resulting query thus captures the exact behavior difference, producing alarms on the vulnerable snapshot and not on the patched snapshot.

Together, these capabilities let QLCoder synthesize a semantically precise CodeQL query that can be reused for regression testing, variant analysis, or patch validation. We now elaborate on the detailed design and implementation of QLCoder.

## 3  QLCODER

At a high level, QLCoder operates inside a repository-aware iterative refinement loop (Figure 3). In each iteration, the agent proposes a candidate CodeQL query, a CodeQL-based validator executes and scores it on both the vulnerable and patched versions of the repository, and the agent uses the validation feedback to propose targeted repairs. The loop terminates successfully when the validator accepts a query, or fails after a fixed iteration budget. In this section, we elaborate the major design components that make the loop effective.

### 3.1 PROBLEM STATEMENT

The task of vulnerability detection is generally framed as a taint analysis task, where the goal of a *query* is to find dataflow paths from a *source* (e.g., an API endpoint accepting user input) to a *sink* (e.g., a database write) that lack proper *sanitization* (e.g., filtering malicious data).

We formalize the *Vulnerability Query Synthesis* problem as follows. Assume as input a vulnerable project version $P_{\text{vuln}}$, its fixed version $P_{\text{fixed}}$, and a textual CVE description (commonly available in open vulnerability reports). Let us assume we have inter-procedural dataflow program graphs for each code version: $G_{\text{vuln}} = (V_{\text{vuln}}, E_{\text{vuln}})$ and $G_{\text{fixed}} = (V_{\text{fixed}}, E_{\text{fixed}})$. Let $\Delta P$ denote the source-level patch between $P_{\text{vuln}}$ and $P_{\text{fixed}}$. We represent the patch in the dataflow-graph domain as a patch subgraph $\Delta G = (\Delta V, \Delta E)$, where $\Delta V$ is the set of graph nodes that correspond to the modified program snippets.

A *vulnerability path query* $Q$ evaluated on a graph $G$ returns a set of dataflow paths, denoted as $\Pi = [\![Q]\!](G)$. We write each path $\pi \in \Pi$ as $\pi = \langle v_1, \ldots, v_k \rangle$, where each $v_i \in V$ is a node in the dataflow graph $G$. Consecutive nodes $(v_i, v_{i+1})$ should be either connected by an existing edge in $E$, or an *additional taint step* specified in the query $Q$, to compensate for missing edges via dataflow graph construction. Specifically, we call $v_1$ the *source* of path $\pi$ and $v_k$ the *sink* of $\pi$.

**Synthesis task.** We aim to synthesize a query $Q$ from the vulnerability report satisfying the following requirements:

1. **Well-formedness.** $Q$ is syntactically valid (based on the latest CodeQL syntax) and can be executed on the target CodeQL infrastructure (e.g., dataflow graphs) without runtime errors.
2. **Vulnerability detection.** $Q$ generates at least one path $\pi$ in the vulnerable version that traverses the patched region:
$$\exists \pi \in [\![Q]\!](G_{\text{vuln}}) \quad \text{such that} \quad \pi \cap \Delta V \neq \emptyset.$$
3. **Fix discrimination.** $Q$ does not report the vulnerability in the fixed version. Concretely, no path reported on the fixed version should traverse the patched locations:
$$\forall \pi \in [\![Q]\!](G_{\text{fixed}}), \quad \text{we have} \quad \pi \cap \Delta V = \emptyset.$$

In other words, the synthesized query must be executable, must witness the vulnerability in the vulnerable version via a path that uses code touched by the fix, and must not attribute the same (patched) behavior in the fixed version. When only the well-formedness condition is satisfied, we say that the query $Q$ is valid (denoted as $\text{valid}(Q)$); when all the conditions are satisfied, the query $Q$ is successful (denoted as $\text{success}(Q; P_{\text{vuln}}, P_{\text{fixed}})$). Note that these criteria may admit potentially false positive paths in both versions. It might be possible to consider additional constraints regarding precision, but it might further complicate synthesis. In practice, we find most queries synthesized by QLCoder already have high precision.

### 3.2 DESIGN OF QLCODER

Concretely, QLCoder proceeds in an iterative refinement loop indexed by $i = [0, 1, \ldots]$. Via prompting, the LLM agent-based synthesizer first proposes an initial candidate query $Q_0$. For each iteration $i$, the validator evaluates $Q_i$ and produces a feedback report. We consider synthesis successful at iteration $i$ iff $\text{success}(Q_i; P_{\text{vuln}}, P_{\text{fixed}})$ holds; in that case the loop terminates and $Q_i$ is returned. Otherwise, the synthesizer analyzes the feedback and the previous candidate $Q_i$, and produces the next query candidate $Q_{i+1}$. The loop stops successfully when $\text{success}(\cdot)$ is achieved or fails once $i$ reaches the pre-configured limit $N$ (in our implementation $N = 10$). The remainder of the design focuses on two aspects: 1) how the agentic synthesizer performs synthesis, and 2) how the validator generates and communicates feedback. We elaborate on both below.

**Agentic synthesizer.** In each iteration $i$, the LLM-based agentic synthesizer runs an inner *conversation loop* of up to $M$ turns. In each turn, the agent either performs internal reasoning or issues a tool call by emitting a JSON-formatted action. When a tool call succeeds, the tool returns a JSON-formatted response that is appended to the conversation history. Conversation histories are kept local to the current refinement iteration (i.e., not carried over between iterations) to keep context compact and relevant. In practice, we set $M = 50$, i.e., the agent may interact with tools up to 50 times before generating a candidate query for validation.

Two design choices are critical for the effectiveness of this loop: 1) the *initial prompt* that initializes and constrains the agent's behavior, and 2) the *toolbox* of callable tools, each exposed by a custom

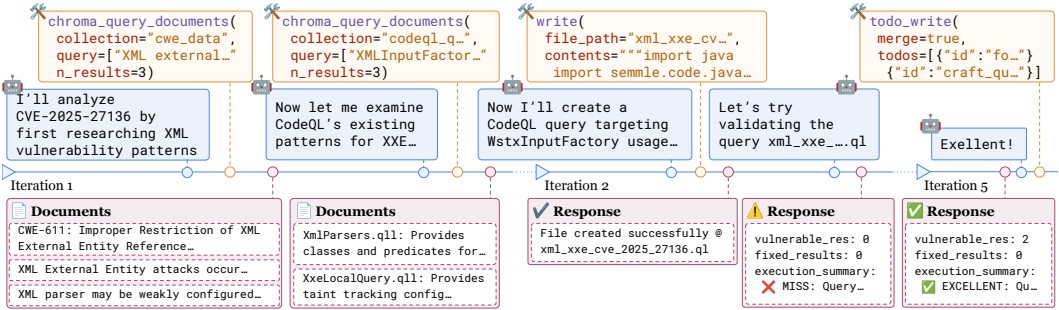

Figure 4: Illustration of example traces of conversation during the synthesis of the query in the motivating example (Figure 2). LLM-agent may think, invoke tools that are available in the toolbox, and receive responses from the MCP servers.

Model Context Protocol (MCP) server. We refer to the combined problem of designing these items as *Context Engineering* (discussed in Section 3.3).

**CodeQL-Based Validator.** The validator compiles and executes each candidate query against the vulnerable and fixed versions and returns a concise, structured feedback report that is used to drive refinement (Figure 3). The vulnerable and patched versions are automatically derived from patch commit hash, which is in the CVE metadata. The report contains: (i) CodeQL compilation results, (ii) execution counts (matches on vulnerable and fixed graphs), (iii) recall and coverage statistics, (iv) concrete counterexample traces and hit locations, and (v) a prioritized set of next-step recommendations (e.g., add qualifiers, synthesize sanitizer checks, or expand taint steps) that are programmatically generated via a template.

## 3.3 CONTEXT ENGINEERING FOR AGENTIC SYNTHESIZER

The primary goal of context engineering is to expose the LLM-based agent to the most *precise* amount of information: enough for the agent to make progress, but not so much that the LLM is confused or the cost explodes. As illustrated in Figure 3, QLCoder relies on two primary MCP servers to provide demand-driven, structured information to the agent: a retrieval-augmented vector database and a CodeQL Language Server interface. We show example traces of conversation loop in Figure 4 and describe the available tools below.

**Initial prompt.** Each refinement iteration begins with an *initial prompt* that kicks-off the agentic conversation loop. The initial prompt in the first iteration contains a query skeleton for reference (See § D.1 for an example). In subsequent iterations, the prompt contains a summary of the synthesis goal and constraints, the previous candidate query $Q_{i-1}$, and the validator feedback report. Concretely, the initial prompt emphasizes: (i) the success predicate (see $\text{Success}(\cdot)$), (ii) concrete counterexamples from previous feedback, and (iii) an explicit list of callable tools and their purpose.

**Vector database.** We use a retrieval-augmented vector database (ChromaDB MCP server in our implementation) to store large reference corpora without polluting the LLM prompt. The database is pre-populated with (i) vulnerability analysis notes and diffs, (ii) Common Weaknesses Enumeration (CWE) definitions, (iii) same-version CodeQL API documentation, (iv) curated CodeQL sample queries, and (v) small abstract syntax tree (AST) snippets extracted from the target repository. We manually filtered the patch diffs files to be only related to the given CVE and automated removing non-Java files that couldn't be analyzed by CodeQL or were unrelated to the CVE pattern, such as binary files. Afterwards, ASTs of the related patch files were extracted. During a conversation loop, the agent issues compact retrieval queries (e.g., to fetch example CodeQL queries related to the CWE) and receives ranked documents or snippets on demand.

In practice, we may populate our RAG database with tens of thousands of documents. Even with this large corpus, we observe that the LLM-agent reliably retrieves exactly the kinds of artifacts it needs: CodeQL sample queries that inspire overall query structure, small AST snippets that suggest the precise syntactic navigation, and vulnerability writeups or diff excerpts that help discriminate buggy from patched behavior. These demand-driven lookups let the agent gather high-quality information without loading the main prompt with large reference corpora.

**CodeQL language server.** We expose the CodeQL Language Server (GitHub, 2025i) through a MCP server that the agent can call for precise syntax-aware guidance. Importantly, we developed our own CodeQL Language Server client and MCP server that ensures syntactic validity (especially for the given CodeQL version) during query generation. The LLM agent's MCP client makes the tool call which is received by the CodeQL MCP server. The MCP server forwards tool calls, such as `complete(file, loc, char)`, `diagnostics(file)`, and `definition(file, loc, char)`, to the underlying CodeQL process and returns JSON-serializable responses. Tools such as `completion` help the agent fill query templates and discover correct API or AST names, while `diagnostics` reveal compile or linter errors (e.g., unknown predicate names) that guide mutation. Appendix C shows the full specification and example request and response schemas.

## 3.4 DESIGN DECISIONS

We discuss several alternative designs that we considered but found ineffective in practice. Allowing the agent unrestricted access to compile-and-run CodeQL via MCP led to severe performance degradation: compilation and full execution are expensive operations that the LLM soon overused, so we instead expose only lightweight diagnostics during the conversation and defer full compile-and-run to the end of each iteration. Permitting web search for vulnerability patterns or snippets similarly proved problematic. It is both costly and easy for the agent to rely on web lookups, which quickly pollutes the working context and degrades synthesis quality. See Appendix B for more details on using web search. Equipping the agent with an extensive set of heterogeneous tools led to confusion and poor tool-selection behavior; in contrast, a small, well-scoped toolbox yields more reliable actions. Finally, retaining full conversation histories across refinement iterations induced context rot and ballooning prompt sizes, so we keep histories local to each iteration. Overall, our current design is a pragmatic trade-off that balances cost, responsiveness, and synthesis effectiveness.

## 4 EVALUATION

We aim to answer the following research questions through our empirical evaluation:

- **RQ 1**: For how many CVEs can QLCoder successfully generate queries?
- **RQ 2**: How useful is each component of QLCoder?
- **RQ 3:** How does the choice of base agent framework affect QLCoder's effectiveness?
- **RQ 4:** How scalable is QLCoder?

## 4.1 EXPERIMENTAL SETUP

We develop QLCoder on top of the Claude Code framework (Anthropic, 2025a) and use Claude Sonnet 4 for all our experiments. For agent baselines, we select Codex with GPT-5 (minimal reasoning) and Gemini CLI with Gemini 2.5 Flash. For each CVE and agent baseline, we use a maximum of 10 iterations ($N = 10$). For static analysis baselines, we select IRIS Li et al. (2025b) and CodeQL (version 2.22.2) query suites. Experiments were run on machines with the following specifications: for the gpt-oss agent baseline we used a machine with an Intel Xeon Gold 6338 2.00 GHz CPU, 10x NVIDIA H100 PCIe, and 1 TB RAM. For all other experiments we used an Intel Xeon Gold 6248 2.50GHz CPU, four GeForce RTX 2080 Ti GPUs, and 750GB RAM.

**Dataset.** We used CWE-Bench-Java (Li et al., 2025b) and its latest update, which added new CVEs from 2025. We were able to successfully build and use 111 (out of 120) Java CVEs evaluated in IRIS (Li et al., 2025b), and 65 (out of 91) 2025 CVEs. Each sample in CWE-Bench-Java comes with the CVE metadata and fix commit information associated with the bug.

## 4.2 EVALUATION METRICS

Besides `valid(Q)` and `success(Q; P_{vuln}, P_{fixed})` from Section 3.1, we use the following terms and metrics when evaluating QLCoder and baselines on the problem of vulnerability query synthesis:

$$\text{Rec}(Q) = \mathbb{1}[\exists \pi \in [\![Q]\!](G_{\text{vuln}}), \pi \cap \Delta V \neq \emptyset], \quad \text{Prec}(Q) = \frac{|\{\pi \in [\![Q]\!](G_{\text{vuln}}) \mid \pi \cap \Delta V \neq \emptyset\}|}{|[\![Q]\!](G_{\text{vuln}})|},$$

$$\text{F1}(Q) = 2 \cdot \frac{\text{Prec}(Q) \cdot \text{Rec}(Q)}{\text{Prec}(Q) + \text{Rec}(Q)}.$$

## 4.3 RQ1: QLCODER EFFECTIVENESS

**QLCoder vs. state-of-the-art QL.** Table 1 shows QLCoder's overall query synthesis success rate by CWE. Table 2 shows the notable increase in precision of QLCoder over CodeQL and IRIS.

Table 1: QLCoder Query Success by CWE Type.

| CWE Type | Total CVEs | # Success | Success (%) | Avg Precision |
|---|---|---|---|---|
| CWE-022 (Path Traversal) | 48 | 31 | 64.6 | 0.75 |
| CWE-079 (Cross-Site Scripting) | 36 | 18 | 50.0 | 0.621 |
| CWE-094 (Code Injection) | 20 | 12 | 60.0 | 0.606 |
| CWE-078 (OS Command Injection) | 12 | 7 | 58.3 | 0.628 |
| CWE-502 (Deserialization) | 6 | 4 | 66.7 | 0.853 |
| CWE-611 (XXE) | 5 | 3 | 60.0 | 0.657 |
| Other CWEs ($\leq$4 CVEs) | 49 | 19 | 38.8 | 0.504 |
| **Total** | **176** | **94** | **53.4** | **0.631** |

Table 2: Recall Performance Comparison Across Methods (Shared CVEs: 130).

| Method | Recall Rate (%) | Avg Precision | Avg F1 Score |
|---|---|---|---|
| CodeQL | 20.0 | 0.055 | 0.073 |
| IRIS | 35.4 | 0.031 | 0.048 |
| QLCoder | 80.0 | 0.672 | 0.700 |

QLCoder is able to successfully synthesize 53.4% of the CVEs. For half the queries QLCoder correctly synthesizes CodeQL, detects the CVE, and does not report false positives on the fixed version of the CVE's repository. The lack of true positive recall is why CodeQL and IRIS have significantly lower precision. CodeQL's queries are broad, categorized by CWE queries. IRIS generates all of the predicates for potential sources and sinks with CodeQL, and does not generate sanitizer or taint step predicates.

Finally, Figure 5 shows that CodeQL, IRIS, and QLCoder have significantly higher vulnerability recall rates compared to Snyk and SpotBugs. Thus highlighting CodeQL's superior performance compared to other static analysis tools.

**Impact of training cut-off.** We also want to take note that Claude Sonnet 4's training cut-off is March 2025. Table 3 shows that QLCoder performs consistently regardless of CVEs before or after the cut-off period. The CodeQL version, 2.22.2, was released in July 2025. New versions of CodeQL often include analysis improvements and new QL packs (GitHub, 2025f).

Table 3: Tool Performance Before vs After Training Cutoff.

| CVE Period | Total CVEs | # Recall | Success (%) | Avg Precision | Avg F1 Score |
|---|---|---|---|---|---|
| Pre-2025 (2011-2024) | 111 | 64 | 57.7 | 0.676 | 0.702 |
| 2025+ (Post-cutoff) | 65 | 30 | 46.2 | 0.555 | 0.583 |
| **Overall** | **176** | **94** | **53.4** | **0.631** | **0.658** |

## 4.4 RQ2: ABLATION STUDIES

For ablations, we chose 20 CVEs and ran QLCoder with one of the QLCoder components removed (Table 4). The ablation with no tools refers to only running Claude Code with the iterative feedback system. The high recall rate when removing access to the AST cache while lowered recall rates without the LSP server or documentation access show that the LSP and documentation lookup impact the synthesis performance more. We also include QLCoder's performance on the same set of CVEs, and point out its significantly higher query success rate and precision score. Claude Code without tools scored a high recall rate, yet failed to synthesize queries without false positives when executed on the fixed version.

## 4.5 RQ3: STATE OF THE ART AGENT COMPARISON

QLCoder can be transferred to other coding agents. For switching between coding agents, the MCP server configuration format may differ but the MCP server commands and arguments are the same. Coding agents also had different CLI commands used to invoke the agent. For the weaker, open source model we used (gpt-oss:20b), we had to break down our prompts into smaller tasks and increase query writing iterations. We used Gemini CLI with Gemini 2.5 Flash and Gemini 2.5 Pro. We used Codex with GPT-5 minimal reasoning effort, GPT-5 medium reasoning effort, and for trying open source models. We used Codex with open source models due to ease of configuration. See Appendix A for more details on using Codex with open source models. We evaluated their performance on 20 CVEs. We achieved an increase in compilation success for both agents compared

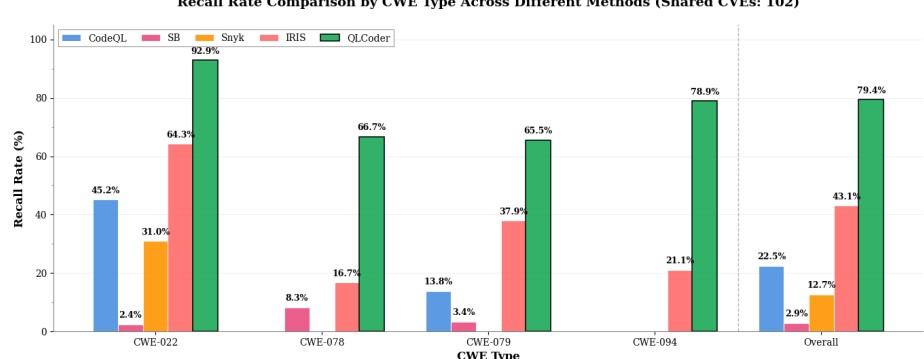

Figure 5: Recall Rate Comparison by CWE Type Across Different Methods (102 CVEs).

Table 4: Ablation Study (out of 20 CVEs).

| Variant | % Successful | Recall Rate | Avg Precision | Avg F1 Score |
|---|---|---|---|---|
| QLCoder | **55%** | **80%** | **0.67** | **0.69** |
| w/o LSP | 25% (−30%) | 55% (−25%) | 0.32 | 0.36 |
| w/o Doc/Ref | 20% (−35%) | 55% (−25%) | 0.32 | 0.36 |
| w/o AST | 25% (−30%) | 80% (±0%) | 0.41 | 0.47 |
| Claude Code | 10% (−45%) | 55% (−25%) | 0.33 | 0.36 |

to using the agents without QLCoder in Table 5. We also achieved an increase in successful queries generated using Codex and GPT-5 medium reasoning with QLCoder. GPT-5 minimal reasoning with QLCoder also had a slight increase compared to not using, though not as large of a gain compared to using medium reasoning effort. Codex with gpt-oss-20b also had a slight increase in successful queries synthesized, compared to without QLCoder. See Appendix A for statistics on common error categories for agent baseline failed queries.

### 4.6 RQ4: SCALABILITY OF QLCODER

To compare costs and runtime of CodeQL query writing with QLCoder, do note that GitHub offers 8 consulting services related to CodeQL query writing to businesses (GitHub, 2025e). For QLCoder, the average time to synthesize each query took 3712 seconds. The average cost for each synthesized query was 2.90 USD. The average input token usage was 708 tokens per query. The average output token usage was 43176 tokens per query. Figure 6 shows how for queries that finish synthesizing in < 3000 seconds, the success rate is around 97%. Thus, one could consider an early stop policy around 3000 seconds. A more detailed breakdown of runtime and cost per CVE can be found in Appendix A.

## 5 DISCUSSION

### 5.1 VERSATILITY OF QLCODER

**CodeQL Versions and Other Languages.** To upgrade the CodeQL version for QLCoder, the system has to fetch CodeQL documentation and queries related to the upgraded version. We used a simple python script to fetch CodeQL documentation web pages and each CodeQL CLI installation comes with its version-adjusted queries. To use QLCoder for other languages supported by CodeQL,

Table 5: LLM-agent baselines' compilation and success rates on 20 CVEs from 2025.

| Agent Baselines | Language Model | w/o QLCoder | | with QLCoder | |
|---|---|---|---|---|---|
| | | Compilation | Successful | Compilation | Successful |
| Gemini CLI | Gemini 2.5 Flash | 19% | 0% | 24% (+5%) | 0% (=) |
| Gemini CLI | Gemini 2.5 Pro | 35% | 0% | 75% (+40%) | 0% (=) |
| Codex | GPT-5, Minimal | 0% | 0% | 24% (+24%) | 5% (+5%) |
| Codex | GPT-5, Medium | 0% | 0% | 55% (+55%) | 20% (+20%) |
| Codex | gpt-oss-20b | 25% | 0% | 35% (+10%) | 5% (+5%) |
| Claude Code | Sonnet 4 | 95% | 10% | 100% (+5%) | 35% (+25%) |

the system has to fetch CodeQL documentation and queries specific to the language, which are categorized in the CLI installation and the site documentation.

**Other SAST tools.** For adapting the system for other engines, Semgrep (Semgrep, 2025) and Snyk (Snyk, 2025) both have officially supported language servers. For engines that don't have language servers, one can create a MCP server that interfaces with calling the engine's commands.

### 5.2 APPLICATIONS OF QLCODER

**Variant Analysis.** A QLCoder synthesized query can be used for variant analysis. By running the known CVE pattern on other repositories, we can find new bugs that match the pattern. CodeQL's VSCode extension already supports running a given query on the top 1000 public repositories or any specified group of repositories (GitHub, 2025c). For example, with one QLCoder synthesized query we reported 2 unknown bugs in 2 different repositories.

**Regression Testing.** In continuous integration pipelines, regression testing is common component to have. One of Github's CI features is its CodeQL code scan Github action (GitHub, 2025a). The action automatically runs CodeQL queries on a given pull request or specified event. Currently CodeQL's provided security queries support only 64 CWEs (GitHub, 2025h). QLCoder is CWE agnostic and given a history of known CVEs for a specific repository, the synthesized queries can be used as regression testing via a CI pipeline.

## 6 RELATED WORK

**LLMs and vulnerability detection.** LLMs have been used extensively for vulnerability detection and repair using techniques such as fine-tuning and prompt engineering (Zhou et al., 2024). LLMs have also been combined with existing program analysis tools for vulnerability detection. The combination of LLMs can be used from vulnerability analysis like IRIS's (Li et al., 2025b) source and sink identification, however IRIS depends on a limited set of CWE templates derived from CodeQL's CWE queries. IRIS also only the LLM for identifying sources and sinks. KNighter synthesizes CSA checkers given a fix commit of a C repository (Yang et al., 2025), however the checkers are written in C which has more available training data. MocQ's uses an LLM to derive a subset DSL of CodeQL and Joern, and then provides a feedback loop to the LLM though prompting via API calls is used rather than an agent with tools and MocQ uses significantly higher iterations, with a max threshold of 1,000 iterations per vulnerability experiment. (Li et al., 2025a).

**LLM agents and tool usage.** SWE-agent pioneered the idea of autonomous LLM agents using tools for software engineering tasks Yang et al. (2024). LSPAI Go et al. (2025), an IDE plugin, uses LSP servers to guide LLM-generated unit tests. Hazel, a live program sketching environment, uses a language server (Blinn et al., 2024) to assist code completions synthesized by LLMs. The Hazel Language Server provides the typing context of a program hole to be filled.

**Low resource LLM code generation.** SPEAC uses ASTs combined with constraint solving to repair LLM-generated code for low resource programming languages (Mora et al., 2024). SPEAC converts a buggy program into an AST and uses a solver to find the minimum set of AST nodes to replace, to satisfy language constraints. MultiPL-T generates datasets for low resource languages by translating high resource language code to the target language and validates translations with LLM generated unit tests (Cassano et al., 2024).

## 7 CONCLUSION AND LIMITATIONS

We present QLCoder, an agentic framework for synthesizing syntactically correct and precise CodeQL queries given known vulnerability patterns. We will also open source our CodeQL LSP MCP server and QLCoder. In future work, we plan to explore efficient ways to synthesize, and to combine our synthesized queries with dynamic analysis tools.

**Limitations.** We omit CVEs where the vulnerability involves non-Java code such as configuration files or other languages. QLCoder can be used with exploit generation to find vulnerabilities that are realized during dynamic execution. For supporting other languages that can be queried by CodeQL, the vector database can be filled with references, documentation, and example queries in other CodeQL supported languages. We also note that Claude Sonnet 4's official training cut-off is March 2025, however the 2025 CVEs evaluated were reported between January to August 2025.

ACKNOWLEDGMENTS

We thank the reviewers for feedback that improved this paper. This research was supported by NSF award CCF #2313010, NSF CISE Graduate Fellowships under Grant No. 2313998, and the Google TPU Builders Program.

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

## A  EVALUATION DETAILS

Table 6 is a more detailed breakdown of the successful query synthesis rate by CWE.

**Runtime and Costs.** Table 7 is a more detailed breakdown of cost and runtime per synthesized CVE query. In table 7, SLOC is the number of source lines of code. SLOC is calculated by using the CLI tool tokei (XAMPPRocky, 2025) on the project checked out at the patch commit and counting the number of lines of code for Java files of the project. Time was derived from recording the timestamps of starting QLCoder and when QLCoder returned a successful query or exhausted 10 iterations of the query. Note that the analysis phase invokes Claude Code once, and the query writing phase invokes Claude Code up to 10 times. Thus for deriving total tokens used, we summed up the following token values found from the following keys in each Claude Code API response (as of September 24, 2025) - `input_tokens`, `cache_creation_input_tokens`, `cache_read_input_tokens`, and `output_tokens`. In Table 7, the Tokens column shows token counts in thousands. For the USD cost value, we added up each Claude Code API response's `total_cost_usd` value in a given CVE synthesis. In Table 7, the USD column shows the total cost of synthesizing the query.

**Error Categories for Agent Baselines' from Table 5.** We derived the error categories for why a query wasn't successful in each of the agent baseline configurations. The categories are: (i) the patched version has vulnerability recall, (ii) the query did not detect any of the patched methods

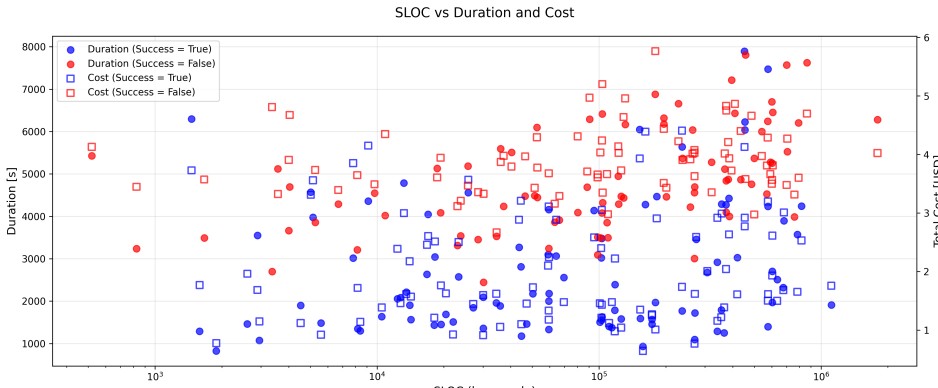

Figure 6: SLOC, Synthesis Duration, and Synthesis Cost Comparison by CVE.

related to the CVE, (iii) the query did have a file recall but failed to alarm on the vulnerable methods in the file, and (iv) the query did not compile. We have also included Figure 7 to visualize each agent baseline configurations' error category count.

**CodeQL Baseline Queries.** The CodeQL baseline queries used are from CodeQL's security QL pack (GitHub, 2025h). We picked the CVEs that have CWEs supported by CodeQL's security queries. Not all of CWEs represented in our evaluation have corresponding CodeQL CWE queries.

**Using Codex with Open Source Models.** As of November 17, 2025, to use Codex with open source models, we used ollama (Ollama, 2025) to download and host the models. Codex has a con-figuration file located at `[home directory]/.codex/config.toml`. For each open source model we wanted to try out with QLCoder, we added a new profile for the model in the configuration file. We also were able to use existing MCP server configurations for Codex, with the open source models. Below is an example of the configuration for using gpt-oss:20b with Codex. Afterwards, to use the profile with Codex we use the command `codex --profile gpt-oss-20b`.

```
[model_providers.ollama]
name = "Ollama"
base_url = "http://localhost:11434/v1"
[profiles.gpt-oss-20b]
model_provider = "ollama"
model = "gpt-oss:20b"
wire_api = "chat"
```

## A.1 EVALUATION LIMITATIONS

**Codex CLI.** Claude Code and Gemini CLI allow users to configure how many max turns an agent can take in a context window. As of 9/23/2025, Codex CLI does not offer this configu-ration. Thus we were not able to force Codex to always take up to 50 max turns each context window.

**IRIS.** The original IRIS evaluation consists of 120 Java projects from CWE-Bench-Java. Many of these projects are old with deprecated dependencies, thus we were only able to build and use 112 of the projects with CodeQL 2.22.2. As of 9/23/2025, IRIS supports 11 CWEs and out of the 65 CVEs from 2025, we were able to use 24 of them with IRIS. When running some of the IRIS queries, the amount of sources and sink predicates in the query led to out of memory errors. This impacted 9 out of the 24 queries, thus we treat those as queries with 0 results and false recall.

Table 6: QLCoder Query Success by CWE Type.

| CWE Type | Total CVEs | # Success | Success (%) | Avg Precision |
|---|---|---|---|---|
| CWE-022 (Path Traversal) | 48 | 31 | 64.6% | 0.750 |
| CWE-079 (Cross-Site Scripting) | 36 | 18 | 50.0% | 0.621 |
| CWE-094 (Code Injection) | 20 | 12 | 60.0% | 0.606 |
| CWE-078 (OS Command Injection) | 12 | 7 | 58.3% | 0.628 |
| CWE-502 (Deserialization) | 6 | 4 | 66.7% | 0.853 |
| CWE-611 (XXE) | 5 | 3 | 60.0% | 0.657 |
| CWE-287 (Authentication) | 4 | 1 | 25.0% | 0.875 |
| CWE-200 (Information Exposure) | 3 | 0 | 0.0% | 0.667 |
| CWE-400 (Resource Consumption) | 3 | 1 | 33.3% | 0.556 |
| CWE-532 (Information Exposure) | 3 | 3 | 100.0% | 0.686 |
| CWE-770 (Resource Exhaustion) | 3 | 1 | 33.3% | 0.444 |
| CWE-020 (Improper Input Validation) | 2 | 2 | 100.0% | 0.650 |
| CWE-089 (SQL Injection) | 2 | 2 | 100.0% | 1.000 |
| CWE-1333 (ReDoS) | 2 | 0 | 0.0% | 0.000 |
| CWE-284 (Access Control) | 2 | 0 | 0.0% | 0.500 |
| CWE-862 (Authorization) | 2 | 0 | 0.0% | 0.000 |
| CWE-918 (SSRF) | 2 | 1 | 50.0% | 0.500 |
| CWE-023 (Relative Path Traversal) | 1 | 1 | 100.0% | 1.000 |
| CWE-044 (Path Equivalence) | 1 | 1 | 100.0% | 0.667 |
| CWE-083 (Improper Neutralization) | 1 | 1 | 100.0% | 0.052 |
| CWE-1325 (Improperly Controlled Memory) | 1 | 0 | 0.0% | 0.000 |
| CWE-164 (Foreign Code) | 1 | 0 | 0.0% | 0.000 |
| CWE-178 (Case Sensitivity) | 1 | 0 | 0.0% | 1.000 |
| CWE-190 (Integer Overflow) | 1 | 0 | 0.0% | 0.000 |
| CWE-264 (Permissions) | 1 | 0 | 0.0% | 0.000 |
| CWE-267 (Privilege Defined) | 1 | 0 | 0.0% | 0.000 |
| CWE-276 (Incorrect Permissions) | 1 | 0 | 0.0% | 1.000 |
| CWE-285 (Improper Authorization) | 1 | 1 | 100.0% | 1.000 |
| CWE-288 (Authentication Bypass) | 1 | 0 | 0.0% | 0.000 |
| CWE-290 (Authentication Bypass) | 1 | 1 | 100.0% | 1.000 |
| CWE-297 (Improper Certificate) | 1 | 1 | 100.0% | 1.000 |
| CWE-312 (Cleartext Storage) | 1 | 0 | 0.0% | 0.000 |
| CWE-327 (Cryptographic Issues) | 1 | 0 | 0.0% | 0.000 |
| CWE-346 (Origin Validation) | 1 | 0 | 0.0% | 0.200 |
| CWE-352 (CSRF) | 1 | 1 | 100.0% | 0.941 |
| CWE-426 (Untrusted Search Path) | 1 | 0 | 0.0% | 0.000 |
| CWE-835 (Infinite Loop) | 1 | 0 | 0.0% | 0.000 |
| CWE-863 (Authorization) | 1 | 1 | 100.0% | 1.000 |
| **Total** | **176** | **94** | **53.4%** | **0.631** |

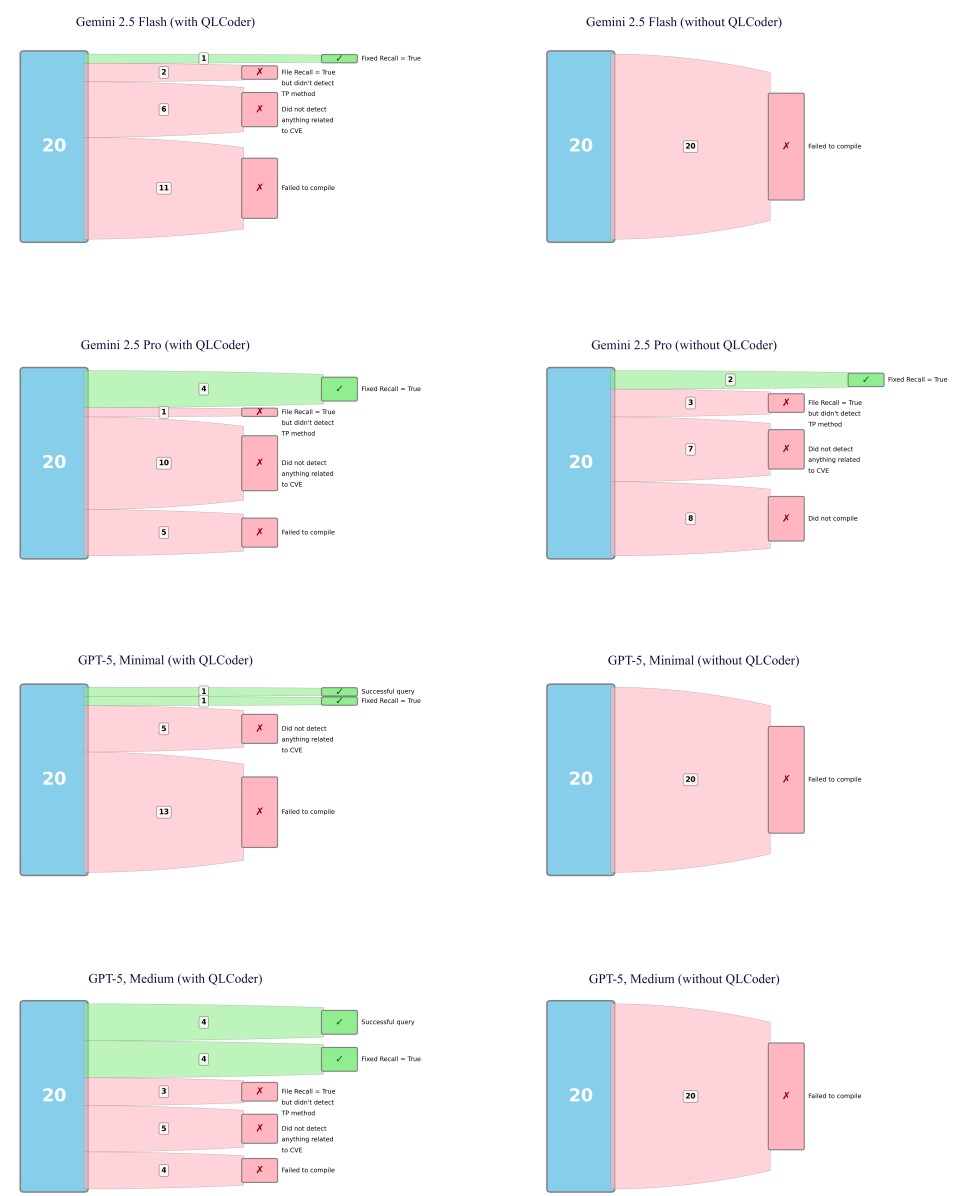

Figure 7: Error Decomposition for Table 5.

Table 7: QLCoder duration, costs, and total tokens used per CVE ID.

| CVE-ID | CWE-ID | Project | SLOC | Time (s) | Tokens (K) | USD |
|---|---|---|---|---|---|---|
| CVE-2025-24970 | 020 | netty | 341.20K | 2916 | 5060.08 | 2.92 |
| CVE-2025-22233 | 020 | spring-framework | 782.76K | 3570 | 2750.30 | 1.65 |
| CVE-2022-23082 | 022 | CureKit | 519 | 5429 | 6415.35 | 4.13 |
| CVE-2022-31194 | 022 | DSpace | 237.97K | 5367 | 5934.22 | 3.91 |
| CVE-2022-31195 | 022 | DSpace | 236.54K | 5636 | 5346.47 | 4.41 |
| CVE-2025-53622 | 022 | DSpace | 373.20K | 4275 | 3052.49 | 2.04 |
| CVE-2018-12036 | 022 | DependencyCheck | 28.53K | 3453 | 5467.16 | 3.36 |
| CVE-2021-29425 | 022 | commons-io | 40.33K | 5511 | 5844.67 | 3.79 |

| CVE-ID | CWE-ID | Project | SLOC | Time (s) | Tokens (K) | USD |
|---|---|---|---|---|---|---|
| CVE-2025-0851 | 022 | djl | 102.97K | 3477 | 4968.76 | 3.05 |
| CVE-2022-26884 | 022 | dolphinscheduler | 90.69K | 6288 | 7767.97 | 4.97 |
| CVE-2022-34662 | 022 | dolphinscheduler | 110.91K | 1402 | 1748.13 | 1.12 |
| CVE-2022-23457 | 022 | esapi-java-legacy | 35.88K | 1888 | 1529.97 | 1.05 |
| CVE-2020-17519 | 022 | flink | 1.11M | 1912 | 3395.07 | 1.76 |
| CVE-2022-26049 | 022 | goomph | 12.77K | 2084 | 2150.50 | 1.46 |
| CVE-2023-41044 | 022 | graylog2-server | 270.29K | 1722 | 2778.67 | 1.62 |
| CVE-2018-17297 | 022 | hutool | 64.12K | 3063 | 4408.37 | 2.88 |
| CVE-2022-22931 | 022 | james-project | 434.53K | 4865 | 6468.51 | 4.40 |
| CVE-2020-8570 | 022 | java | 816.52K | 4238 | 3653.89 | 2.53 |
| CVE-2025-49656 | 022 | jena | 676.42K | 2319 | 2795.36 | 1.68 |
| CVE-2019-0225 | 022 | jspwiki | 59.35K | 2002 | 2183.08 | 1.33 |
| CVE-2022-3782 | 022 | keycloak | 576.48K | 4234 | 2193.97 | 1.50 |
| CVE-2023-35887 | 022 | mina-sshd | 131.18K | 6165 | 7486.34 | 4.96 |
| CVE-2020-35460 | 022 | mpxj | 181.97K | 4468 | 4041.71 | 2.91 |
| CVE-2011-4367 | 022 | myfaces | 263.92K | 6037 | 6228.39 | 4.01 |
| CVE-2022-25842 | 022 | one-java-agent | 5.59K | 1481 | 1118.59 | 0.92 |
| CVE-2018-1002200 | 022 | plexus-archiver | 13.54K | 2213 | 2585.96 | 1.61 |
| CVE-2023-37460 | 022 | plexus-archiver | 12.36K | 2056 | 3958.90 | 2.39 |
| CVE-2022-4244 | 022 | plexus-utils | 23.84K | 3540 | 5521.94 | 3.21 |
| CVE-2018-1000850 | 022 | retrofit | 20.43K | 1688 | 2831.06 | 1.63 |
| CVE-2019-17572 | 022 | rocketmq | 94.98K | 4136 | 3845.71 | 2.58 |
| CVE-2023-34478 | 022 | shiro | 37.14K | 4233 | 6733.75 | 3.98 |
| CVE-2023-46749 | 022 | shiro | 43.62K | 3268 | 4812.75 | 2.89 |
| CVE-2024-23673 | 022 | sling-org-apache-sling-servlets-resolver | 8.41K | 1297 | 1730.85 | 1.14 |
| CVE-2025-46096 | 022 | solon | 125.80K | 1579 | 1695.92 | 1.04 |
| CVE-2016-9177 | 022 | spark | 9.74K | 4547 | 5257.03 | 3.49 |
| CVE-2018-9159 | 022 | spark | 10.87K | 4017 | 6149.27 | 4.35 |
| CVE-2020-5405 | 022 | spring-cloud-config | 22.01K | 1511 | 1414.43 | 0.92 |
| CVE-2020-5410 | 022 | spring-cloud-config | 18.66K | 5131 | 5543.48 | 3.61 |
| CVE-2019-0207 | 022 | tapestry-5 | 157.63K | 932 | 1092.47 | 0.64 |
| CVE-2018-11762 | 022 | tika | 109.76K | 3494 | 4456.12 | 2.98 |
| CVE-2022-32287 | 022 | uima-uimaj | 227.86K | 6658 | 5932.88 | 4.23 |
| CVE-2014-7816 | 022 | undertow | 103.90K | 4320 | 5419.59 | 3.35 |
| CVE-2022-36007 | 022 | venice | 117.70K | 1783 | 1371.73 | 0.98 |
| CVE-2018-12542 | 022 | vertx-web | 34.46K | 1961 | 2487.06 | 1.62 |
| CVE-2019-17640 | 022 | vertx-web | 47.21K | 1462 | 2660.48 | 1.45 |
| CVE-2018-1047 | 022 | wildfly | 486.38K | 4759 | 7287.55 | 4.66 |
| CVE-2023-45277 | 022 | yamcs | 173.18K | 1457 | 2031.62 | 1.27 |
| CVE-2023-45278 | 022 | yamcs | 173.18K | 1572 | 2204.78 | 1.25 |
| CVE-2018-1002202 | 022 | zip4j | 8.18K | 1348 | 2894.83 | 1.72 |
| CVE-2018-1002201 | 022 | zt-zip | 7.80K | 3016 | 5901.48 | 3.85 |
| CVE-2025-1584 | 023 | solon | 118.06K | 2388 | 2999.88 | 2.22 |
| CVE-2025-24813 | 044 | tomcat | 366.29K | 1252 | 2409.97 | 1.39 |
| CVE-2025-26074 | 078 | conductor | 114.24K | 1376 | 2726.99 | 1.48 |
| CVE-2022-20617 | 078 | docker-commons-plugin | 2.89K | 3551 | 2354.26 | 1.68 |
| CVE-2019-10392 | 078 | git-client-plugin | 16.79K | 2630 | 4160.82 | 2.46 |
| CVE-2020-2261 | 078 | perfecto-plugin | 825 | 3238 | 5544.05 | 3.45 |
| CVE-2017-1000487 | 078 | plexus-utils | 23.30K | 2571 | 3938.81 | 2.50 |
| CVE-2023-24422 | 078 | script-security-plugin | 8.15K | 3211 | 6250.12 | 3.65 |
| CVE-2022-25174 | 078 | workflow-cps-global-lib-plugin | 4.53K | 1901 | 1634.79 | 1.12 |
| CVE-2022-25173 | 078 | workflow-cps-plugin | 16.98K | 4048 | 3594.86 | 2.61 |
| CVE-2022-25175 | 078 | workflow-multibranch-plugin | 3.58K | 5120 | 4783.47 | 3.32 |
| CVE-2013-7285 | 078 | xstream | 44.44K | 2809 | 5416.36 | 3.21 |
| CVE-2020-26217 | 078 | xstream | 52.51K | 6096 | 6573.86 | 4.29 |
| CVE-2021-21345 | 078 | xstream | 52.77K | 4443 | 5313.38 | 3.78 |

*Continued on next page*

| CVE-ID | CWE-ID | Project | SLOC | Time (s) | Tokens (K) | USD |
|---|---|---|---|---|---|---|
| CVE-2022-31192 | 079 | DSpace | 236.52K | 1767 | 2638.92 | 1.74 |
| CVE-2016-10006 | 079 | antisamy | 4.01K | 3664 | 6565.33 | 3.91 |
| CVE-2017-14735 | 079 | antisamy | 4.04K | 4696 | 7181.97 | 4.68 |
| CVE-2022-28367 | 079 | antisamy | 5.03K | 4569 | 4505.33 | 3.31 |
| CVE-2022-29577 | 079 | antisamy | 5.14K | 3973 | 5344.35 | 3.56 |
| CVE-2025-47885 | 079 | cloudbees-jenkins-advisor-plugin | 2.60K | 1461 | 3818.47 | 1.96 |
| CVE-2025-2901 | 079 | console | 122.59K | 4287 | 5209.18 | 4.03 |
| CVE-2016-6812 | 079 | cxf | 602.83K | 2700 | 2878.97 | 1.93 |
| CVE-2019-17573 | 079 | cxf | 680.23K | 3897 | 4745.86 | 3.01 |
| CVE-2022-24891 | 079 | esapi-java-legacy | 36.07K | 5593 | 5719.59 | 3.87 |
| CVE-2020-27219 | 079 | hawkbit | 103.46K | 6416 | 6996.26 | 5.20 |
| CVE-2019-10219 | 079 | hibernate-validator | 88.47K | 4691 | 6373.29 | 3.71 |
| CVE-2018-1000129 | 079 | jolokia | 30.21K | 2441 | 5901.65 | 3.33 |
| CVE-2025-32961 | 079 | jpawebapi | 3.37K | 2698 | 7642.30 | 4.81 |
| CVE-2020-13973 | 079 | json-sanitizer | 1.67K | 3489 | 5929.25 | 3.57 |
| CVE-2019-10076 | 079 | jspwiki | 59.38K | 3244 | 5633.10 | 3.68 |
| CVE-2019-10077 | 079 | jspwiki | 59.38K | 1335 | 1950.83 | 1.18 |
| CVE-2019-10078 | 079 | jspwiki | 59.38K | 4160 | 4980.42 | 3.11 |
| CVE-2019-10089 | 079 | jspwiki | 59.55K | 2177 | 3361.70 | 2.24 |
| CVE-2022-46907 | 079 | jspwiki | 58.95K | 3097 | 3359.56 | 2.10 |
| CVE-2023-33962 | 079 | jstachio | 19.33K | 4083 | 7190.32 | 3.94 |
| CVE-2025-23026 | 079 | jte | 23.05K | 3313 | 5277.52 | 3.12 |
| CVE-2014-3656 | 079 | keycloak | 66.27K | 3912 | 5476.51 | 3.29 |
| CVE-2022-1274 | 079 | keycloak | 576.88K | 7472 | 4463.84 | 3.20 |
| CVE-2022-4137 | 079 | keycloak | 576.99K | 1397 | 2976.16 | 1.61 |
| CVE-2022-4361 | 079 | keycloak | 575.34K | 6241 | 7337.30 | 4.29 |
| CVE-2021-44667 | 079 | nacos | 122.11K | 4944 | 5964.66 | 4.14 |
| CVE-2025-32960 | 079 | restapi | 19.41K | 1452 | 3001.09 | 1.76 |
| CVE-2016-5394 | 079 | sling-org-apache-sling-xss | 1.46K | 6294 | 5968.91 | 3.73 |
| CVE-2025-46558 | 079 | syntax-markdown | 2.95K | 1075 | 2051.41 | 1.15 |
| CVE-2023-29201 | 079 | xwiki-commons | 101.47K | 3493 | 7209.06 | 4.03 |
| CVE-2023-29528 | 079 | xwiki-commons | 102.52K | 1555 | 2139.98 | 1.21 |
| CVE-2023-31126 | 079 | xwiki-commons | 103.00K | 1627 | 2454.59 | 1.43 |
| CVE-2023-36471 | 079 | xwiki-commons | 103.17K | 4079 | 6791.06 | 4.24 |
| CVE-2023-37908 | 079 | xwiki-rendering | 51.28K | 4486 | 5594.74 | 3.43 |
| CVE-2020-29204 | 079 | xxl-job | 9.12K | 4361 | 6101.32 | 4.15 |
| CVE-2023-32070 | 083 | xwiki-rendering | 50.36K | 2175 | 2917.20 | 1.73 |
| CVE-2022-45206 | 089 | jeecgboot | 100.87K | 1505 | 2707.01 | 1.46 |
| CVE-2022-4963 | 089 | spring-module-core | 1.59K | 1287 | 3680.27 | 1.77 |
| CVE-2019-0222 | 094 | activemq | 409.42K | 6429 | 7499.98 | 4.86 |
| CVE-2020-11998 | 094 | activemq | 419.39K | 3025 | 2437.65 | 1.61 |
| CVE-2022-42889 | 094 | commons-text | 27.12K | 1842 | 2152.16 | 1.45 |
| CVE-2021-41269 | 094 | cron-utils | 13.21K | 4784 | 4689.80 | 3.00 |
| CVE-2023-49109 | 094 | dolphinscheduler | 153.39K | 1591 | 2369.87 | 1.35 |
| CVE-2023-51770 | 094 | dolphinscheduler | 152.57K | 6049 | 6345.60 | 3.93 |
| CVE-2021-30180 | 094 | dubbo | 179.47K | 1968 | 1735.59 | 1.01 |
| CVE-2021-30181 | 094 | dubbo | 179.12K | 6880 | 8732.56 | 5.76 |
| CVE-2022-44262 | 094 | ff4j | 46.47K | 4480 | 6612.80 | 3.97 |
| CVE-2021-4178 | 094 | kubernetes-client | 69.41K | 2556 | 2116.29 | 1.48 |
| CVE-2025-30067 | 094 | kylin | 395.31K | 7213 | 5382.61 | 3.72 |
| CVE-2023-34468 | 094 | nifi | 864.61K | 7622 | 8168.47 | 4.70 |
| CVE-2023-33246 | 094 | rocketmq | 108.61K | 3850 | 5853.05 | 3.66 |
| CVE-2023-37582 | 094 | rocketmq | 201.07K | 4461 | 5656.22 | 3.43 |
| CVE-2022-46166 | 094 | spring-boot-admin | 18.23K | 3040 | 4401.71 | 2.51 |
| CVE-2022-22947 | 094 | spring-cloud-gateway | 25.77K | 4561 | 3338.67 | 3.57 |
| CVE-2022-22965 | 094 | spring-framework | 705.22K | 5523 | 6182.90 | 4.27 |

| CVE-ID | CWE-ID | Project | SLOC | Time (s) | Tokens (K) | USD |
|---|---|---|---|---|---|---|
| CVE-2018-1260 | 094 | spring-security-oauth | 44.68K | 1176 | 1851.95 | 1.10 |
| CVE-2023-32697 | 094 | sqlite-jdbc | 18.11K | 1436 | 2001.01 | 1.21 |
| CVE-2020-17530 | 094 | struts | 161.73K | 4277 | 7378.37 | 4.39 |
| CVE-2025-2240 | 1325 | smallrye-fault-tolerance | 34.56K | 3527 | 4291.18 | 2.67 |
| CVE-2025-48058 | 1333 | powsybl-core | 269.01K | 4560 | 6698.58 | 4.08 |
| CVE-2025-48059 | 1333 | powsybl-core | 269.35K | 4691 | 6438.45 | 4.03 |
| CVE-2025-30177 | 164 | camel | 1.80M | 6280 | 7088.64 | 4.02 |
| CVE-2025-24399 | 178 | oic-auth-plugin | 5.26K | 3860 | 5949.46 | 3.73 |
| CVE-2025-52520 | 190 | tomcat | 381.92K | 4871 | 6579.21 | 4.21 |
| CVE-2025-26795 | 200 | iotdb | 759.13K | 3989 | 5053.95 | 3.31 |
| CVE-2025-54380 | 200 | opencast | 257.74K | 4220 | 6120.22 | 3.92 |
| CVE-2025-22227 | 200 | reactor-netty | 80.17K | 4088 | 7084.78 | 4.31 |
| CVE-2014-3576 | 264 | activemq | 321.48K | 5276 | 5521.34 | 3.35 |
| CVE-2025-23015 | 267 | cassandra | 500.47K | 5368 | 4546.67 | 2.98 |
| CVE-2025-24790 | 276 | snowflake-jdbc | 98.59K | 3512 | 5499.07 | 3.66 |
| CVE-2025-48734 | 284 | commons-beanutils | 25.69K | 5183 | 5848.92 | 3.47 |
| CVE-2025-1391 | 284 | keycloak | 591.27K | 5277 | 6025.32 | 3.67 |
| CVE-2025-53106 | 285 | graylog2-server | 356.87K | 4287 | 5071.88 | 2.99 |
| CVE-2023-51982 | 287 | crate | 540.71K | 5998 | 6567.43 | 3.97 |
| CVE-2025-0604 | 287 | keycloak | 570.32K | 4526 | 6785.14 | 3.81 |
| CVE-2025-3910 | 287 | keycloak | 600.36K | 6701 | 5630.84 | 3.50 |
| CVE-2025-22228 | 287 | spring-security | 273.92K | 3460 | 4215.36 | 2.59 |
| CVE-2025-49125 | 288 | tomcat | 374.12K | 4096 | 7034.93 | 4.75 |
| CVE-2025-22223 | 290 | spring-security | 307.39K | 2675 | 3320.81 | 2.01 |
| CVE-2025-3501 | 297 | keycloak | 603.77K | 1971 | 4672.52 | 2.61 |
| CVE-2025-53103 | 312 | junit-framework | 125.80K | 4471 | 5648.19 | 3.73 |
| CVE-2025-27508 | 327 | emissary | 63.19K | 3865 | 5184.05 | 3.16 |
| CVE-2025-7365 | 346 | keycloak | 606.26K | 6454 | 5955.44 | 3.81 |
| CVE-2025-31723 | 352 | simple-queue-plugin | 1.89K | 824 | 1248.24 | 0.78 |
| CVE-2025-48795 | 400 | cxf | 700.41K | 7568 | 5211.09 | 3.48 |
| CVE-2025-25193 | 400 | netty | 341.19K | 1290 | 1896.56 | 1.16 |
| CVE-2025-53506 | 400 | tomcat | 373.95K | 4838 | 7602.65 | 4.83 |
| CVE-2025-24789 | 426 | snowflake-jdbc | 98.59K | 3089 | 5808.05 | 3.61 |
| CVE-2025-27522 | 502 | inlong | 456.66K | 7805 | 5149.58 | 3.58 |
| CVE-2025-27526 | 502 | inlong | 453.50K | 6227 | 4254.62 | 2.77 |
| CVE-2025-27528 | 502 | inlong | 453.50K | 6036 | 4720.41 | 2.92 |
| CVE-2025-27531 | 502 | inlong | 452.17K | 7894 | 6402.01 | 4.13 |
| CVE-2025-27818 | 502 | kafka | 792.42K | 6204 | 5508.74 | 3.60 |
| CVE-2025-47771 | 502 | powsybl-core | 269.40K | 1094 | 1366.41 | 0.77 |
| CVE-2025-48955 | 532 | para | 30.08K | 2094 | 2722.48 | 1.60 |
| CVE-2025-49009 | 532 | para | 30.08K | 1357 | 1624.77 | 0.91 |
| CVE-2025-27496 | 532 | snowflake-jdbc | 102.30K | 3024 | 4348.54 | 2.40 |
| CVE-2025-53621 | 611 | DSpace | 386.12K | 3998 | 6407.57 | 3.93 |
| CVE-2025-52888 | 611 | allure2 | 14.21K | 1564 | 2836.58 | 1.57 |
| CVE-2025-27136 | 611 | local-s3 | 14.03K | 1904 | 3533.41 | 2.18 |
| CVE-2025-47293 | 611 | powsybl-core | 269.25K | 3006 | 6040.41 | 3.63 |
| CVE-2025-4641 | 611 | webdrivermanager | 10.50K | 1633 | 2034.01 | 1.39 |
| CVE-2025-48976 | 770 | commons-fileupload | 6.69K | 4289 | 5690.52 | 3.39 |
| CVE-2025-32959 | 770 | cuba | 384.90K | 4424 | 4115.30 | 2.63 |
| CVE-2025-48988 | 770 | tomcat | 368.45K | 5115 | 6289.47 | 4.02 |
| CVE-2025-27497 | 835 | OpenDJ | 601.66K | 5256 | 5809.20 | 3.56 |
| CVE-2025-31720 | 862 | jenkins | 195.74K | 6319 | 5288.47 | 3.51 |
| CVE-2025-31721 | 862 | jenkins | 195.81K | 6179 | 7173.86 | 4.44 |
| CVE-2025-27427 | 863 | activemq-artemis | 634.70K | 2506 | 2809.01 | 1.50 |
| CVE-2023-51441 | 918 | axis-axis1-java | 128.99K | 4432 | 8246.73 | 4.64 |
| CVE-2024-48944 | 918 | kylin | 355.34K | 1791 | 1932.82 | 1.22 |

| CVE-ID | CWE-ID Project | SLOC | Time (s) | Tokens (K) | USD |
|--------|----------------|------|----------|------------|-----|
| **Average** | | **209.44K** | **3712** | **4614.56** | **2.90** |

# B  DESIGN DECISIONS

## B.1  WEB SEARCH TOOL USAGE

In section 3.4 we discuss how free web search for vulnerability patterns was a tool we decided not to use. Free web search had less constraints in the context window. Web search lookups could fail due to web requests potentially dropping and we did not want to risk using up the context window from reliability issues. Web search also led to the agent searching for things that potentially weren't related to the CVE. We only want the agent to analyze the CVE description to better inform the vulnerability pattern. Although web search could open up the agent to more resources, we also were concerned about using too many tokens in the context window thus preferred cacheing CVE entries from the NVD database instead of performing a web search for every analysis call for the same CVE. Below we have provided transcripts for the vulnerability analysis phase for the same CVE (CVE-2025-27136) - the first example is where Claude Code's Web fetch tool (Anthropic, 2025b) and Web search tool (Anthropic, 2025c) are enabled, thus the agent can use these tools for free web search. The second example is where Web fetch and Web search are not enabled. The second example's configuration of having no free web search is what we use in QLCoder- we cache the CVE entries from the NVD website and we cache CWE data from MITRE in the vector database.

```
1  Turn 5 - Assistant
2  Tool: WebSearch
3  Query: CVE-2025-27136 vulnerability details XML external entity XXE
4  Tokens: Input: 5 | Cache Read: 29,596 | Cache Create: 267 | Output: 67
5  Result: Web search results for query: "CVE-2025-27136 vulnerability
6  details XML external entity XXE"
7
8  I'll search for information about the CVE-2025-27136 vulnerability and its
9  relation to XML external entity (XXE) issues.
10
11 Links: [{"title":"LocalS3 CreateBucketConfiguration Endpoint XML External Entity
12 (XXE) Injection . CVE-2025-27136 . GitHub Advisory Database . GitHub",
13 "url":"https://github.com/advisories/GHSA-g6wm-2v64-wq36"},
14 {"title":"CVE-2025-27136 Impact, Exploitability, and Mitigation Steps | Wiz",
15 "url":"https://www.wiz.io/vulnerability-database/cve/cve-2025-27136"},
16 {"title":"NVD - CVE-2025-27136",
17 "url":"https://nvd.nist.gov/vuln/detail/CVE-2025-27136"},
18 {"title":"XML External Entity (XXE) Processing | OWASP Foundation",
19 "url":"https://owasp.org/www-community/vulnerabilities/XML_External_Entity_(XXE)_
20 Processing"},{"title":"What is XXE (XML External Entity) | Examples & Prevention |
21 Imperva","url":"https://www.imperva.com/learn/application-security/
22 xxe-xml-external-entity/"},{"title":"CVE - Search Results",
23 "url":"https://cve.mitre.org/cgi-bin/cvekey.cgi?keyword=xxe"},
24 {"title":"What is XXE (XML external entity) injection? Tutorial & Examples
25 | Web Security Academy","url":"https://portswigger.net/web-security/xxe"},
26 {"title":"XXE Complete Guide: Impact, Examples, and Prevention | HackerOne",
27 "url":"https://www.hackerone.com/knowledge-center/
28 xxe-complete-guide-impact-examples-and-prevention"},{"title":"XML External
29 Entity (XXE)","url":"https://www.invicti.com/learn/xml-external-entity-xxe"},
30 {"title":"XML External Entity Prevention - OWASP Cheat Sheet Series","url":
31 "https://cheatsheetseries.owasp.org/cheatsheets/
32 XML_External_Entity_Prevention_Cheat_Sheet.html"}]
33
34 Turn 6 - Assistant
35 Tool: WebSearch
36 Query: XML external entity XXE CodeQL detection patterns Java XMLInputFactory
37 Tokens: Input: 4 | Cache Read: 30,998 | Cache Create: 267 | Output: 64
38 Result: Web search results for query: "XML external entity XXE CodeQL detection patterns Java
       XMLInputFactory"
39
40 I'll search for information about XML external entity (XXE) CodeQL detection patterns for Java
        XMLInputFactory.
41
42 Links: [{"title":"XML External Entity Prevention - OWASP Cheat Sheet Series",
43 "url":"https://cheatsheetseries.owasp.org/cheatsheets/XML_External_Entity_Prevention
```

```
44 _Cheat_Sheet.html"},{"title":"Resolving XML external entity in user-controlled data - CodeQL
        query help documentation","url":"https://codeql.github.com/codeql-query-help/java/java-
        xxe/"},
45 {"title":"Java false positive: XXE via XMLInputFactory . Issue #7199 . github/codeql","url":"
        https://github.com/github/codeql/issues/7199"},
46 {"title":"XML Entity Expansion in Java | SecureFlag Security Knowledge Base","url":"https://
        knowledge-base.secureflag.com/vulnerabilities/xml_injection/xml_entity_expansion_java.
        html"},
47 {"title":"Detecting, Fixing, and Defending Against XXE Attacks in Python and Java | by Rohit
        kumar | Medium","url":"https://rohitcoder.medium.com/comprehensive-guide-detecting-fixing
        -and-defending-against-xxe-attacks-in-python-and-java-e78691b4b918"},{"title":"Prevent
        XML External Entity Vulnerabilities for Java - xxe","url":"https://semgrep.dev/docs/cheat
        -sheets/java-xxe"},{"title":"how to fix 'Disable XML external entity (XXE) processing'
        vulnerabilities in java - Stack Overflow","url":"https://stackoverflow.com/questions
        /56777287/how-to-fix-disable-xml-external-entity-xxe-processing-vulnerabilities-in-jav
        "},{"title":"XXE vulnerability detection | Moderne Docs","url":"https://docs.moderne.io/
        openrewrite-advanced-program-analysis/security/xxe/"},{"title":"How to prevent XML
        external entity attack (XXE attack) - Mkyong.com","url":"https://mkyong.com/java/how-to-
        prevent-xml-external-entity-attack-xxe-attack/"},{"title":"xml - Java XXE vulnerability -
         Information Security Stack Exchange","url":"https://security.stackexchange.com/questions
        /260956/java-xxe-vulnerability"}]
48
49
50 Turn 7 - Assistant
51 Tool: WebSearch
52 Query: CodeQL XXE taint tracking Java XML parsing sources sinks sanitizers
53 Tokens: Input: 3 | Cache Read: 31,265 | Cache Create: 1,354 | Output: 66
54 Result: Web search results for query: "CodeQL XXE taint tracking Java XML parsing sources
        sinks sanitizers"
55
56 I'll search for information about CodeQL XXE taint tracking in Java XML parsing, including
        sources, sinks, and sanitizers.
57
58 Links: [{"title":"Practical Introduction to CodeQL :: jorgectf - blog","url":"https://jorgectf
        .github.io/blog/post/practical-codeql-introduction/"},{"title":"Resolving XML external
        entity in user-controlled data - CodeQL query help documentation","url":"https://codeql.
        github.com/codeql-query-help/java/java-xxe/"},{"title":"Using flow state for precise data
         flow analysis - CodeQL","url":"https://codeql.github.com/docs/codeql-language-guides/
        using-flow-labels-for-precise-data-flow-analysis/"},
59 {"title":"CodeQL 2.16.4 (2024-03-11) CodeQL","url":"https://codeql.github.com/docs/codeql-
        overview/codeql-changelog/codeql-cli-2.16.4/"},{"title":"Analyzing data flow in Java and
        Kotlin  CodeQL","url":"https://codeql.github.com/docs/codeql-language-guides/analyzing-
        data-flow-in-java/"},{"title":"RealworldCTF 2024 Protected-by-Java-SE
60 Writeup","url":"https://intrigus.org/research/2025/05/07/realworld-ctf-2024-using-codeql-to-
        find-bugs-in-codeql/"},{"title":"New dataflow API for writing custom CodeQL queries -
        GitHub
61 Changelog","url":"https://github.blog/changelog/2023-08-14-new-dataflow-api-for-writing-custom
        -codeql-queries/"},
62 {"title":"TaintTracking - CodeQL - GitHub","url":"https://codeql.github.com/codeql-standard-
        libraries/javascript/semmle/javascript/dataflow/TaintTracking.qll/module.TaintTracking.
        html"},{"title":"How codeql deal with java's interface invoke in Global Taint Tracking .
        github/codeql . Discussion #7363","url":"https://github.com/github/codeql/discussions
        /7363"},{"title":"CodeQL 2.11.0 (2022-09-28) - CodeQL","url":"https://codeql.github.com/
        docs/codeql-overview/codeql-changelog/codeql-cli-2.11.0/"}]}
```

Listing 1: Web search and Web fetch tools are enabled for Claude Code to use during the vulnerability analysis phase. Note: we truncated the analysis and are only showing the tool usage, tokens used, and the web links searched.

```
1 Turn 4 - Assistant
2 Tool:chroma_get_documents(collection_name:nist_cve_2025_brief,where:
3 {"cve_id": "CVE-2025-27136"})
4 Tokens: Input: 6 | Cache Read: 28,074 | Cache Create: 340 | Output: 105
5
6 Turn 6 - Assistant
7 Tool: chroma_query_documents(collection_name: cwe_data, query_texts:
8 ['CWE-611', 'XML External Entity', 'XXE injection'],n_results: 3)
9 Tokens: Input: 5 | Cache Read: 29,031 | Cache Create: 340 | Output: 121
```

Listing 2: Web search and Web fetch tools are disabled for Claude Code to use during the vulnerability analysis phase. Claude Code can use cached data related to vulnerability patterns stored in the vector database. We have only provided the tool calls and token usage.

## C  CODEQL LANGUAGE SERVER VIA MCP

The following are the MCP tool specifications and example usage for our custom CodeQL LSP client, wrapped as an MCP server. The MCP server code can be found at `https://github.com/neuralprogram/codeql-lsp-mcp`.

### C.1  TOOL SPECIFICATIONS

**codeql_complete**  Provides code completions at a specific position in a CodeQL file. Supports pagination for large completion lists and trigger character-based completions.

**Inputs:**

- `file_uri` (string): The URI of the CodeQL file
- `line` (number): Line number (0-based)
- `character` (number): Character position in the line (0-based)
- `trigger_character` (string, optional): Optional trigger character (e.g., ".", "::")
- `limit` (number, optional): Maximum number of completion items to return (default: 50)
- `offset` (number, optional): Starting position for pagination (default: 0)

**Returns:** `CompletionList` with pagination metadata containing completion items, each with label, kind, documentation, and text edit information.

Example usage:

```
{
  "tool": "codeql_complete",
  "arguments": {
    "file_uri": "file:///workspace/security-query.ql",
    "line": 5,
    "character": 12,
    "trigger_character": ".",
    "limit": 25
  }
}
```

**codeql_hover**  Retrieves hover information (documentation, type information) at a specific position. Provides rich markdown documentation for CodeQL predicates, classes, and modules.

**Inputs:**

- `file_uri` (string): The URI of the CodeQL file
- `line` (number): Line number (0-based)
- `character` (number): Character position in the line (0-based)

**Returns:** `Hover | null` containing documentation content in markdown or plain text format, with optional range highlighting.

Example usage:

```
{
  "tool": "codeql_hover",
  "arguments": {
    "file_uri": "file:///workspace/security-query.ql",
    "line": 8,
    "character": 15
  }
}
```

**codeql_definition**   Navigates to the definition location for a symbol at a specific position. Supports both single definitions and multiple definition locations.

**Inputs:**

- `file_uri` (string): The URI of the CodeQL file
- `line` (number): Line number (0-based)
- `character` (number): Character position in the line (0-based)

**Returns:** `Location | Location[] | null` containing URI and range information for definition locations.

Example usage:

```
{
  "tool": "codeql_definition",
  "arguments": {
    "file_uri": "file:///workspace/security-query.ql",
    "line": 12,
    "character": 8
  }
}
```

**codeql_references**   Finds all references to a symbol at a specific position across the workspace. Includes both usage references and declaration references.

**Inputs:**

- `file_uri` (string): The URI of the CodeQL file
- `line` (number): Line number (0-based)
- `character` (number): Character position in the line (0-based)

**Returns:** `Location[] | null` containing an array of all reference locations with URI and range information.

Example usage:

```
{
  "tool": "codeql_references",
  "arguments": {
    "file_uri": "file:///workspace/security-query.ql",
    "line": 6,
    "character": 20
  }
}
```

**codeql_diagnostics**   Retrieves diagnostics (errors, warnings, information messages) for a CodeQL file. Provides real-time syntax and semantic analysis results.

**Inputs:**

- `file_uri` (string): The URI of the CodeQL file

**Returns:** `Diagnostic[]` containing an array of diagnostic objects with severity, message, range, and optional related information.

Example usage:

```
{
  "tool": "codeql_diagnostics",
  "arguments": {
```

```
    "file_uri": "file:///workspace/security-query.ql"
  }
}
```

**codeql_format**   Formats a CodeQL file or a specific selection within the file according to CodeQL style guidelines.

**Inputs:**

- `file_uri` (string): The URI of the CodeQL file
- `range` (Range, optional): Optional range to format with start and end positions

**Returns:** `TextEdit[]` containing an array of text edits that describe the formatting changes to be applied.

Example usage:

```
{
  "tool": "codeql_format",
  "arguments": {
    "file_uri": "file:///workspace/security-query.ql",
    "range": {
      "start": { "line": 10, "character": 0 },
      "end": { "line": 25, "character": 0 }
    }
  }
}
```

**codeql_update_file**   Updates the content of an open CodeQL file in the language server. This allows for dynamic content modification and analysis of unsaved changes.

**Inputs:**

- `file_uri` (string): The URI of the CodeQL file
- `content` (string): The new complete content of the file

**Returns:** `string` containing a success confirmation message.

Example usage:

```
{
  "tool": "codeql_update_file",
  "arguments": {
    "file_uri": "file:///workspace/security-query.ql",
    "content": "import cpp\n\nfrom Function f\nwhere f.hasName(\"strcpy\")\nselect f,
  }
}
```

## D   CODEQL QUERIES

### D.1   CODEQL QUERY STRUCTURE TEMPLATE

The template below is given to the LLM agent at the start of the iterative query synthesis task. The prompt instructs the LLM to use the AST nodes, along with the CodeQL LSP and CodeQL references in the vector database, to fill in this template. The prompt also takes note to find similar queries related to the given CVE's vulnerability.

```
1 /**
2  * @name [Vulnerability Name based on analysis]
3  * @description [Description derived from the vulnerability pattern]
```

```
4   * @problem.severity error
5   * @security-severity [score based on severity]
6   * @precision high
7   * @tags security
8   * @kind path-problem
9   * @id [unique-id]
10  */
11  import java
12  import semmle.code.java.frameworks.Networking
13  import semmle.code.java.dataflow.DataFlow
14  import semmle.code.java.dataflow.FlowSources
15  import semmle.code.java.dataflow.TaintTracking
16  private import semmle.code.java.dataflow.ExternalFlow
17
18  class Source extends DataFlow::Node {
19    Source() {
20      exists([AST node type from analysis] |
21        /* Fill based on AST patterns for sources identified in Phase 1 & 2
        */
22        and this.asExpr() = [appropriate mapping]
23      )
24    }
25  }
26
27  class Sink extends DataFlow::Node {
28    Sink() {
29      exists([AST node type] |
30        /* Fill based on AST patterns for sinks */
31        and this.asExpr() = [appropriate mapping]
32      ) or
33      exists([Alternative AST pattern] |
34        /* Additional sink patterns from analysis */
35        and [appropriate condition]
36      )
37    }
38  }
39
40  class Sanitizer extends DataFlow::Node {
41    Sanitizer() {
42      exists([AST node type for sanitizers] |
43        /* Fill based on sanitizer patterns from Phase 1 & 2 */
44      )
45    }
46  }
47
48  module MyPathConfig implements DataFlow::ConfigSig {
49    predicate isSource(DataFlow::Node source) {
50      source instanceof Source
51    }
52
53    predicate isSink(DataFlow::Node sink) {
54      sink instanceof Sink
55    }
56
57    predicate isBarrier(DataFlow::Node sanitizer) {
58      sanitizer instanceof Sanitizer
59    }
60
61    predicate isAdditionalFlowStep(DataFlow::Node n1, DataFlow::Node n2) {{
62      /* Fill based on additional taint steps from analysis */
63    }}
64  }}
65
66  module MyPathFlow = TaintTracking::Global<MyPathConfig>;
67  import MyPathFlow::PathGraph
```

```
68
69 from
70   MyPathFlow::PathNode source,
71   MyPathFlow::PathNode sink
72 where
73   MyPathFlow::flowPath(source, sink)
74 select
75   sink.getNode(),
76   source,
77   sink,
78   "[Alert_message_based_on_vulnerability]",
79   source.getNode(),
80   "[source_description]"
```

## D.2 ITERATIVE FEEDBACK EXAMPLE

The following is an excerpt from the prompt used in a new context window that instructs the agent to improve the last iteration's query. The feedback is derived from automatically running synthesized queries against the fixed and vulnerable versions of the CVE's repository, and comparing the query's results against the changed methods in the CVE fix commit diff. The feedback also includes any successful method hits, false positive method hits, and which fixed methods have not been detected by the query.

```
1 ## Objective
2 Refine the CodeQL query based on previous iteration feedback to improve
     vulnerability detection.
3
4 ## Previous Iteration Feedback
5 ## Iteration 2 Results
6
7 ## Previous Query (Iteration 2)
8 ```ql
9 /**
10  * @name AntiSamy XSS Bypass in Style Tags
11  * @description Detects potential XSS vulnerabilities where AntiSamy HTML
      sanitization fails to properly filter malicious style tags with
     event handlers
12  * @problem.severity error
13  * @security-severity 6.1
14  * @precision high
15  * @tags security
16  * @kind path-problem
17  * @id java/antisamy-style-xss-bypass
18  */
19 import java
20 import semmle.code.java.dataflow.DataFlow
21 import semmle.code.java.dataflow.TaintTracking
22
23 class MaliciousHtmlInput extends DataFlow::Node {
24   MaliciousHtmlInput() {
25     exists(StringLiteral sl |
26       this.asExpr() = sl and
27       (
28         sl.getValue().matches("%<style%onload%>%") or
29         sl.getValue().matches("%<style%onclick%>%") or
30         sl.getValue().matches("%<style%onerror%>%") or
31         sl.getValue().matches("%<style%onmouseover%>%") or
32         sl.getValue().matches("%style%alert%") or
33         sl.getValue().matches("%<style%javascript%") or
34         sl.getValue().matches("%<style%expression%")
35       )
36     ) or
37     exists(Parameter p |
38       this.asParameter() = p and
```

```
39          (
40            p.getName().toLowerCase().matches("%html%") or
41            p.getName().toLowerCase().matches("%input%") or
42            p.getName().toLowerCase().matches("%content%")
43          )
44      )
45    }
46  }
47
48  class AntiSamyProcessing extends DataFlow::Node {
49    AntiSamyProcessing() {
50      exists(MethodCall mc |
51        this.asExpr() = mc and
52        mc.getMethod().hasName("scan") and
53        mc.getMethod().getDeclaringType()(RefType)
54        .hasQualifiedName("org.owasp.validator.html", "AntiSamy")
55      ) or
56      exists(MethodCall mc |
57        this.asExpr() = mc and
58        mc.getMethod().hasName("getCleanHTML") and
59        mc.getQualifier().getType().(RefType)
60        .hasQualifiedName("org.owasp.validator.html", "CleanResults")
61      )
62    }
63  }
64
65  class PotentialXssOutput extends DataFlow::Node {
66    PotentialXssOutput() {
67      exists(MethodCall mc |
68        this.asExpr() = mc and
69        (
70          mc.getMethod().hasName("contains") and
71          exists(StringLiteral sl |
72            mc.getAnArgument() = sl and
73            (
74              sl.getValue().matches("%alert%") or
75              sl.getValue().matches("%script%") or
76              sl.getValue().matches("%javascript%") or
77              sl.getValue().matches("%onload%") or
78              sl.getValue().matches("%onclick%")
79            )
80          )
81        )
82      ) or
83      exists(MethodCall mc |
84        this.asExpr() = mc and
85        mc.getMethod().hasName("getCleanHTML") and
86        mc.getQualifier().getType().(RefType)
87        .hasQualifiedName("org.owasp.validator.html", "CleanResults")
88      )
89    }
90  }
91
92  class ImproperSanitizationBarrier extends DataFlow::Node {
93    ImproperSanitizationBarrier() {
94      exists(MethodCall mc |
95        this.asExpr() = mc and
96        mc.getMethod().hasName("assertTrue") and
97        exists(LogNotExpr lne |
98          mc.getAnArgument() = lne and
99          lne.getExpr().(MethodCall).getMethod().hasName("contains")
100         )
101     )
102   }
103 }
```

```
104
105 module AntiSamyXssConfig implements DataFlow::ConfigSig {
106   predicate isSource(DataFlow::Node source) {
107     source instanceof MaliciousHtmlInput
108   }
109
110   predicate isSink(DataFlow::Node sink) {
111     sink instanceof PotentialXssOutput
112   }
113
114   predicate isBarrier(DataFlow::Node sanitizer) {
115     sanitizer instanceof ImproperSanitizationBarrier
116   }
117
118   predicate isAdditionalFlowStep(DataFlow::Node n1, DataFlow::Node n2) {
119     exists(MethodCall mc |
120       n1.asExpr() = mc.getQualifier() and
121       n2.asExpr() = mc and
122       mc.getMethod().hasName("scan") and
123       mc.getMethod().getDeclaringType().(RefType)
124       .hasQualifiedName("org.owasp.validator.html", "AntiSamy")
125     ) or
126     exists(MethodCall mc |
127       n1.asExpr() = mc.getQualifier() and
128       n2.asExpr() = mc and
129       mc.getMethod().hasName("getCleanHTML")
130     )
131   }
132 }
133
134 module AntiSamyXssFlow = TaintTracking::Global<AntiSamyXssConfig>;
135 import AntiSamyXssFlow::PathGraph
136
137 from
138   AntiSamyXssFlow::PathNode source,
139   AntiSamyXssFlow::PathNode sink
140 where
141   AntiSamyXssFlow::flowPath(source, sink)
142 select
143   sink.getNode(),
144   source,
145   sink,
146   "Potential XSS vulnerability: HTML input with malicious style tags may
      bypass AntiSamy sanitization",
147   source.getNode(),
148   "malicious HTML input"
149 ```
150
151 ## Compilation Results
152 COMPILATION SUCCESS: Query syntax is valid
153
154 ## Execution Results
155 ## Query Evaluation Summary (Iteration 2)
156 Results: Vulnerable=8, Fixed=8
157 Method Recall: Vulnerable=True, Fixed=True
158 True Positive Methods: Vulnerable=2, Fixed=2
159 Coverage: 1/1 target methods
160 PARTIAL: Query hits targets but has false positives in fixed version
161 Method location format is path/to/hit/file.java:[Class of hit method]:[
      Hit method]
162 Successfully targeted methods:
163   -src/main/java/org/owasp/validator/html/scan/MagicSAXFilter.java:
      MagicSAXFilter:startElement
164 False positives (hits in fixed version):
```

```
165    - src/main/java/org/owasp/validator/html/scan/MagicSAXFilter.java:
       MagicSAXFilter:startElement
166
167 ## Detailed Evaluation Analysis
168
169 **Method Coverage**: 1/1 target methods
170 **File Coverage**: 1/1 target files
171 **Successfully targeted files**:
172    - MagicSAXFilter.java
173
174 **Successfully targeted methods**:
175    - startElement
176
177 **False positives (hit in fixed version)**:
178    - startElement
179
180 **Evaluation Summary**:
181 - Vulnerable DB File Recall: True
182 - Fixed DB File Recall: True
183 - Vulnerable DB Method Recall: True
184 - Fixed DB Method Recall: True
185 - Total Query Results: Vulnerable=8, Fixed=8
186 - Code Flow Paths: Vulnerable=8, Fixed=8
187
188 ## Next Steps
189 We want vulnerable DB method recall and we don't want fixed DB method
       recall!
190 **Priority**: Reduce false positives by adding more specific conditions
       to avoid hitting the methods listed above.
```

### D.3   QLCODER SYNTHESIZED QUERY EXAMPLES

Below are examples of successful synthesized queries. The queries successfully find $> 0$ true positive method hits on the vulnerable CodeQL database of the CVE's source code, and no false positive method hits on the fixed version's CodeQL database. For reference queries to compare with, CWE queries can be found on the official CodeQL repository (GitHub, 2025h;g)

**CVE-2025-27136, CWE-611 - Improper Restriction of XML External Entity Reference**

```
1  /**
2   * @name XML External Entity vulnerability in WstxInputFactory without
        secure configuration
3   * @description WstxInputFactory used in XmlFactory without disabling DTD
         support and external entities allows XXE attacks
4   * @problem.severity error
5   * @security-severity 9.1
6   * @precision high
7   * @tags security
8   * @kind path-problem
9   * @id java/wstxinputfactory-xxe
10  */
11
12 import java
13 import semmle.code.java.dataflow.DataFlow
14 import semmle.code.java.dataflow.FlowSources
15 import semmle.code.java.dataflow.TaintTracking
16 private import semmle.code.java.dataflow.ExternalFlow
17
18 class WstxInputFactoryCreation extends DataFlow::Node {
19   WstxInputFactoryCreation() {
20     exists(MethodCall mc |
21       mc.getMethod().hasQualifiedName("com.ctc.wstx.stax", "
       WstxInputFactory", "newInstance") or
```

```
22       mc.getMethod().hasQualifiedName("com.ctc.wstx.stax", "
      WstxInputFactory", "newFactory")
23      |
24        this.asExpr() = mc
25      ) or
26      exists(ClassInstanceExpr cie |
27        cie.getConstructedType().hasQualifiedName("com.ctc.wstx.stax", "
      WstxInputFactory") and
28        this.asExpr() = cie
29      ) or
30      // Include variable access to WstxInputFactory instances (like "input
      " parameter)
31      exists(Variable v, VarAccess va |
32        v.getType().(RefType).hasQualifiedName("com.ctc.wstx.stax", "
      WstxInputFactory") and
33        va.getVariable() = v and
34        this.asExpr() = va
35      )
36   }
37 }
38
39 class UnsafeXmlFactoryUsage extends DataFlow::Node {
40   UnsafeXmlFactoryUsage() {
41     exists(ClassInstanceExpr xmlFactoryCall |
42       // XmlFactory constructor with WstxInputFactory parameter
43       xmlFactoryCall.getConstructedType()
44       .hasQualifiedName("com.fasterxml.jackson.dataformat.xml", "
      XmlFactory") and
45       xmlFactoryCall.getArgument(0) = this.asExpr()
46     ) or
47     exists(ClassInstanceExpr xmlMapperCall, ClassInstanceExpr
      xmlFactoryCall |
48       // XmlMapper constructor using XmlFactory with WstxInputFactory
49       xmlMapperCall.getConstructedType()
50       .hasQualifiedName("com.fasterxml.jackson.dataformat.xml", "
      XmlMapper") and
51       xmlFactoryCall.getConstructedType()
52       .hasQualifiedName("com.fasterxml.jackson.dataformat.xml", "
      XmlFactory") and
53       xmlMapperCall.getArgument(0) = xmlFactoryCall and
54       xmlFactoryCall.getArgument(0) = this.asExpr()
55     )
56   }
57 }
58
59 class WstxInputFactorySanitizer extends DataFlow::Node {
60   WstxInputFactorySanitizer() {
61     exists(MethodCall setPropertyCall, VarAccess factoryVar |
62       setPropertyCall.getMethod().hasQualifiedName("javax.xml.stream", "
      XMLInputFactory", "setProperty") and
63       setPropertyCall.getQualifier() = factoryVar and
64       (
65         // DTD support disabled
66         (exists(Field f |
67           setPropertyCall.getArgument(0) = f.getAnAccess() and
68           f.hasName("SUPPORT_DTD") and
69           f.getDeclaringType().hasQualifiedName("javax.xml.stream", "
      XMLInputFactory")
70         ) and
71         exists(Field f |
72           setPropertyCall.getArgument(1) = f.getAnAccess() and
73           f.hasName("FALSE") and
74           f.getDeclaringType().hasQualifiedName("java.lang", "Boolean")
75         )) or
76         // External entities disabled
```

```
77          (exists(Field f |
78            setPropertyCall.getArgument(0) = f.getAnAccess() and
79            f.hasName("IS_SUPPORTING_EXTERNAL_ENTITIES") and
80            f.getDeclaringType().hasQualifiedName("javax.xml.stream", "
    XMLInputFactory")
81          ) and
82          exists(Field f |
83            setPropertyCall.getArgument(1) = f.getAnAccess() and
84            f.hasName("FALSE") and
85            f.getDeclaringType().hasQualifiedName("java.lang", "Boolean")
86          ))
87        ) and
88        this.asExpr() = factoryVar
89      )
90    }
91 }
92
93 module WstxInputFactoryFlowConfig implements DataFlow::ConfigSig {
94    predicate isSource(DataFlow::Node source) {
95      source instanceof WstxInputFactoryCreation
96    }
97
98    predicate isSink(DataFlow::Node sink) {
99      sink instanceof UnsafeXmlFactoryUsage
100   }
101
102   predicate isBarrier(DataFlow::Node sanitizer) {
103     sanitizer instanceof WstxInputFactorySanitizer
104   }
105
106   predicate isAdditionalFlowStep(DataFlow::Node n1, DataFlow::Node n2) {
107     // WstxInputFactory passed to XmlFactory constructor
108     exists(ClassInstanceExpr xmlFactoryCall |
109       xmlFactoryCall.getConstructedType()
110       .hasQualifiedName("com.fasterxml.jackson.dataformat.xml", "
    XmlFactory") and
111       xmlFactoryCall.getArgument(0) = n1.asExpr() and
112       n2.asExpr() = xmlFactoryCall
113     ) or
114     // XmlFactory passed to XmlMapper constructor
115     exists(ClassInstanceExpr xmlMapperCall |
116       xmlMapperCall.getConstructedType()
117       .hasQualifiedName("com.fasterxml.jackson.dataformat.xml", "
    XmlMapper") and
118       xmlMapperCall.getArgument(0) = n1.asExpr() and
119       n2.asExpr() = xmlMapperCall
120     )
121   }
122 }
123
124 module WstxInputFactoryFlow = TaintTracking::Global<
     WstxInputFactoryFlowConfig>;
125 import WstxInputFactoryFlow::PathGraph
126
127 from
128   WstxInputFactoryFlow::PathNode source,
129   WstxInputFactoryFlow::PathNode sink
130 where
131   WstxInputFactoryFlow::flowPath(source, sink)
132 select
133   sink.getNode(),
134   source,
135   sink,
136   "WstxInputFactory used without secure configuration flows to XML parser
     , allowing XXE attacks",
```

```
137    source.getNode(),
138    "WstxInputFactory␣usage"
```

### CVE-2025-0851, CWE-22 - Path Traversal

```
1  /**
2   * @name Archive path traversal vulnerability (ZipSlip) – CVE-2025-0851
3   * @description Archive entries with path traversal sequences can write
        files outside the intended extraction directory
4   * @problem.severity error
5   * @security-severity 9.8
6   * @precision high
7   * @tags security
8   * @kind path-problem
9   * @id java/archive-path-traversal-cve-2025-0851
10  */
11
12 import java
13 import semmle.code.java.dataflow.DataFlow
14 import semmle.code.java.dataflow.TaintTracking
15
16 /**
17  * Sources: Archive entry names from ZipEntry.getName() and
        TarArchiveEntry.getName()
18  */
19 class ArchiveEntryNameSource extends DataFlow::Node {
20   ArchiveEntryNameSource() {
21     exists(MethodCall mc |
22       mc.getMethod().getName() = "getName" and    (mc.getMethod().
     getDeclaringType().hasQualifiedName("java.util.zip", "ZipEntry") or
     mc.getMethod()
23             .getDeclaringType()        .hasQualifiedName("org.apache.
     commons.compress.archivers.tar", "TarArchiveEntry")
24       ) and
25       this.asExpr() = mc
26     )
27   }
28 }
29
30 /**
31  * Sinks: Path resolution operations that lead to file creation
32  */
33 class PathCreationSink extends DataFlow::Node {
34   PathCreationSink() {
35     // Arguments to Path.resolve() calls
36     exists(MethodCall resolveCall |
37       resolveCall.getMethod().getName() = "resolve" and
38       resolveCall.getMethod().getDeclaringType()
39       .hasQualifiedName("java.nio.file", "Path") and
40       this.asExpr() = resolveCall.getAnArgument()
41     )
42     or
43     // Arguments to file creation operations
44     exists(MethodCall fileOp |
45       (
46         fileOp.getMethod().getName() = "createDirectories" or
47         fileOp.getMethod().getName() = "newOutputStream" or
48         fileOp.getMethod().getName() = "write" or
49         fileOp.getMethod().getName() = "copy"
50       ) and
51       fileOp.getMethod().getDeclaringType().hasQualifiedName("java.nio.
     file", "Files") and
52       this.asExpr() = fileOp.getAnArgument()
53     )
54   }
```

```qll
55  }
56
57  /**
58   * Sanitizers: Proper validation that prevents path traversal
59   */
60  class PathTraversalSanitizer extends DataFlow::Node {
61    PathTraversalSanitizer() {
62      // The validateArchiveEntry method call that properly validates paths
63      // This blocks flow after the validation call is made
64      exists(MethodCall validateCall |
65        validateCall.getMethod().getName() = "validateArchiveEntry" and
66        (
67          // Any variable assigned from validateArchiveEntry call result
68          exists(Variable v |
69            this.asExpr() = v.getAnAccess() and
70            exists(AssignExpr assign |
71              assign.getDest() = v.getAnAccess() and
72              assign.getRhs() = validateCall
73            )
74          )
75          or
76          // Variables passed through validateArchiveEntry calls
77          this.asExpr() = validateCall.getAnArgument() and
78          exists(ExprStmt stmt | stmt.getExpr() = validateCall)
79        )
80      )
81      or
82      // Proper ".." validation with exception throwing (complete pattern)
83      exists(MethodCall containsCall, IfStmt ifStmt, ThrowStmt throwStmt |
84        containsCall.getMethod().getName() = "contains" and
85        containsCall.getAnArgument().(StringLiteral).getValue() = ".." and
86        ifStmt.getCondition().getAChildExpr*() = containsCall and
87        ifStmt.getThen().getAChild*() = throwStmt and
88        this.asExpr() = containsCall.getQualifier()
89      )
90      or
91      // Path normalization combined with startsWith validation
92      exists(MethodCall normalizeCall, MethodCall startsWithCall |
93        normalizeCall.getMethod().getName() = "normalize" and
94        normalizeCall.getMethod().getDeclaringType()
95        .hasQualifiedName("java.nio.file", "Path") and
96        startsWithCall.getMethod().getName() = "startsWith" and
97        startsWithCall.getMethod().getDeclaringType().
98        hasQualifiedName("java.nio.file", "Path") and
99        DataFlow::localFlow(DataFlow::exprNode(normalizeCall),
100         DataFlow::exprNode(startsWithCall.getQualifier())) and
101        this.asExpr() = normalizeCall.getQualifier()
102     )
103   }
104 }
105
106 /**
107  * Additional predicate to detect validation barriers at method level
108  */
109 predicate hasValidationCall(Callable method) {
110   exists(MethodCall validateCall |
111     validateCall.getEnclosingCallable() = method and
112     validateCall.getMethod().getName() = "validateArchiveEntry"
113   )
114 }
115
116 module PathTraversalConfig implements DataFlow::ConfigSig {
117   predicate isSource(DataFlow::Node source) {
118     source instanceof ArchiveEntryNameSource
119   }
```

```
120
121   predicate isSink(DataFlow::Node sink) {
122     sink instanceof PathCreationSink
123   }
124
125   predicate isBarrier(DataFlow::Node sanitizer) {
126     sanitizer instanceof PathTraversalSanitizer
127   }
128
129   predicate isBarrierIn(DataFlow::Node node) {
130     // Barrier at method entry if method contains validateArchiveEntry
        call
131     node instanceof DataFlow::ParameterNode and
132     hasValidationCall(node.getEnclosingCallable())
133   }
134
135   predicate isAdditionalFlowStep(DataFlow::Node n1, DataFlow::Node n2) {
136     // Flow through variable assignments and declarations
137     exists(LocalVariableDeclExpr decl |
138       decl.getInit() = n1.asExpr() and
139       n2.asExpr() = decl.getVariable().getAnAccess()
140     )
141     or
142     exists(AssignExpr assign |
143       assign.getRhs() = n1.asExpr() and
144       n2.asExpr() = assign.getDest()
145     )
146     or
147     // Flow through string manipulation methods that preserve taint
148     exists(MethodCall mc |
149       mc.getAnArgument() = n1.asExpr() and
150       n2.asExpr() = mc and
151       (
152         mc.getMethod().getName() = "removeLeadingFileSeparator" or
153         mc.getMethod().getName() = "trim" or
154         mc.getMethod().getName() = "toString" or
155         mc.getMethod().getName() = "substring"
156       )
157     )
158     or
159     // Flow through Path operations
160     exists(MethodCall pathOp |
161       pathOp.getAnArgument() = n1.asExpr() and
162       n2.asExpr() = pathOp and
163       pathOp.getMethod().getName() = "resolve" and
164       pathOp.getMethod().getDeclaringType()
165       .hasQualifiedName("java.nio.file", "Path")
166     )
167   }
168 }
169
170 module PathTraversalFlow = TaintTracking::Global<PathTraversalConfig>;
171
172 import PathTraversalFlow::PathGraph
173
174 from PathTraversalFlow::PathNode source, PathTraversalFlow::PathNode sink
175 where
176   PathTraversalFlow::flowPath(source, sink) and
177   // Focus on the specific vulnerable files and methods
178   (
179     source.getNode().getEnclosingCallable().getDeclaringType()
180     .hasName("TarUtils") or
181     source.getNode().getEnclosingCallable().getDeclaringType()
182     .hasName("ZipUtils")
183   ) and
```

```
184    (
185      source.getNode().getEnclosingCallable().getName() = "untar" or
186      source.getNode().getEnclosingCallable().getName() = "unzip"
187    ) and
188    // Only report flows where validation is NOT properly done
189    not hasValidationCall(source.getNode().getEnclosingCallable())
190 select sink.getNode(), source, sink,
191    "Archive_entry_name_from_$@_flows_to_file_system_operation_without_
       proper_path_traversal_validation,_allowing_ZipSlip_attack.",
192    source.getNode(), "archive_entry_name"
```

**CVE-2025-27528, CWE-502 - Deserialization of Untrusted Data**

```
1  /**
2   * @name MySQL JDBC URL parameter injection vulnerability
3   * @description Detects MySQL JDBC URLs with dangerous bracket parameters
       that bypass inadequate filtering in vulnerable code
4   * @problem.severity error
5   * @security-severity 8.8
6   * @precision high
7   * @tags security
8   * @kind path-problem
9   * @id java/mysql-jdbc-url-injection
10  */
11
12 import java
13 import semmle.code.java.dataflow.DataFlow
14 import semmle.code.java.dataflow.TaintTracking
15
16 class MySQLDangerousBracketUrlSource extends DataFlow::Node {
17   MySQLDangerousBracketUrlSource() {
18     // String literals with dangerous MySQL parameters in bracket
       notation
19     exists(StringLiteral lit |
20       lit.getValue().matches("*mysql*") and
21       lit.getValue().matches("*[*]*") and
22       (
23         lit.getValue().matches("*allowLoadLocalInfile*") or
24         lit.getValue().matches("*allowUrlInLocalInfile*") or
25         lit.getValue().matches("*autoDeserialize*") or
26         lit.getValue().matches("*allowPublicKeyRetrieval*") or
27         lit.getValue().matches("*serverTimezone*") or
28         lit.getValue().matches("*user*") or
29         lit.getValue().matches("*password*")
30       ) and
31       this.asExpr() = lit
32     )
33     or
34     // Parameters to filterSensitive method that may contain dangerous
       bracket content
35     exists(Method m, Parameter p |
36       m.hasName("filterSensitive") and
37       m.getDeclaringType().getName() = "MySQLSensitiveUrlUtils" and
38       p = m.getAParameter() and
39       this.asParameter() = p
40     )
41   }
42 }
43
44 class VulnerableCodePatternSink extends DataFlow::Node {
45   VulnerableCodePatternSink() {
46     // The vulnerability: calls to filterSensitive in vulnerable code
       patterns
47     exists(Method m, MethodCall filterCall |
48       m.hasName("filterSensitive") and
```

```
49        m.getDeclaringType().getName() = "MySQLSensitiveUrlUtils" and
50        filterCall.getMethod() = m and
51        this.asExpr() = filterCall and
52        // Key vulnerability condition: this code exists where
      filterSensitiveKeyByBracket method is NOT available
53        // In the vulnerable version, filterSensitiveKeyByBracket doesn't
      exist
54        not exists(Method bracketMethod |
55          bracketMethod.hasName("filterSensitiveKeyByBracket") and
56          bracketMethod.getDeclaringType().getName() = "
      MySQLSensitiveUrlUtils" and
57          bracketMethod.getDeclaringType() = m.getDeclaringType()
58        )
59      )
60      or
61      // Additional sink: method calls that use the result of inadequate
      filtering
62      exists(MethodCall mc, MethodCall filterCall |
63        filterCall.getMethod().hasName("filterSensitive") and
64        filterCall.getMethod().getDeclaringType().getName() = "
      MySQLSensitiveUrlUtils" and
65        DataFlow::localFlow(DataFlow::exprNode(filterCall), DataFlow::
      exprNode(mc.getArgument(_))) and
66        this.asExpr() = mc and
67        // Only vulnerable if no proper bracket filtering exists in the
      same class
68        not exists(Method bracketMethod |
69          bracketMethod.hasName("filterSensitiveKeyByBracket") and
70          bracketMethod.getDeclaringType().getName() = "
      MySQLSensitiveUrlUtils" and
71          bracketMethod.getDeclaringType() = filterCall.getMethod().
      getDeclaringType()
72        )
73      )
74    }
75 }
76
77 class ProperBracketFilteringSanitizer extends DataFlow::Node {
78    ProperBracketFilteringSanitizer() {
79      // The proper bracket-based sanitization method (present only in
      fixed version)
80      exists(MethodCall mc |
81        mc.getMethod().hasName("filterSensitiveKeyByBracket") and
82        mc.getMethod().getDeclaringType().getName() = "
      MySQLSensitiveUrlUtils" and
83        this.asExpr() = mc
84      )
85    }
86 }
87
88 module MySQLJDBCUrlInjectionConfig implements DataFlow::ConfigSig {
89    predicate isSource(DataFlow::Node source) {
90      source instanceof MySQLDangerousBracketUrlSource
91    }
92
93    predicate isSink(DataFlow::Node sink) {
94      sink instanceof VulnerableCodePatternSink
95    }
96
97    predicate isBarrier(DataFlow::Node sanitizer) {
98      sanitizer instanceof ProperBracketFilteringSanitizer
99    }
100
101    predicate isAdditionalFlowStep(DataFlow::Node n1, DataFlow::Node n2) {
102      // Flow through string concatenation operations
```

```
103    exists(AddExpr addExpr |
104      n1.asExpr() = addExpr.getLeftOperand() and
105      n2.asExpr() = addExpr
106    )
107    or
108    exists(AddExpr addExpr |
109      n1.asExpr() = addExpr.getRightOperand() and
110      n2.asExpr() = addExpr
111    )
112    or
113    // Flow through variable assignments
114    exists(Assignment assign |
115      n1.asExpr() = assign.getSource() and
116      n2.asExpr() = assign.getDest()
117    )
118    or
119    // Flow through return statements
120    exists(ReturnStmt ret |
121      n1.asExpr() = ret.getResult() and
122      n2.asParameter() = ret.getEnclosingCallable().getAParameter()
123    )
124  }
125 }
126
127 module MySQLJDBCUrlInjectionFlow = TaintTracking::Global<
       MySQLJDBCUrlInjectionConfig>;
128
129 import MySQLJDBCUrlInjectionFlow::PathGraph
130
131 from MySQLJDBCUrlInjectionFlow::PathNode source,
       MySQLJDBCUrlInjectionFlow::PathNode sink
132 where MySQLJDBCUrlInjectionFlow::flowPath(source, sink)
133 select sink.getNode(), source, sink,
134   "MySQL JDBC URL with dangerous bracket parameters flows to vulnerable
        filtering logic at $@ that lacks proper bracket-based sanitization",
135   source.getNode(), "dangerous URL source"
```

## D.4  AST EXTRACTION QUERY

Given a fix diff, QLCoder automatically parses the changed methods and files, and inserts them into an AST pretty printing query template. Below is an example of the AST extraction query used for CVE-2014-7816.

```
1 /**
2  * @name Expressions and statements for CVE-2014-7816 changed code areas
3  * @description Extract expressions and statements from vulnerability fix
       areas
4  * @id java/expr-stmt-diff-CVE_2014_7816
5  * @kind problem
6  * @problem.severity recommendation
7  */
8
9 import java
10
11 from Element e, Location l
12 where
13   l = e.getLocation() and ((l.getFile().getBaseName() = "
      PathSeparatorHandler.java"
14   and l.getStartLine() >= 1 and l.getEndLine() <= 100) or
15
16   (l.getFile().getBaseName() = "URLDecodingHandler.java"
17   and l.getStartLine() >= 17 and l.getEndLine() <= 128)
18
19   or (l.getFile().getBaseName() = "ResourceHandler.java"
```

```
20    and l.getStartLine() >= 158 and l.getEndLine() <= 172)
21
22    or (l.getFile().getBaseName() = "io.undertow.server.handlers.builder.
        HandlerBuilder" and l.getStartLine() >= 17
23    and l.getEndLine() <= 29)
24
25    or (l.getFile().getBaseName() = "DefaultServlet.java"
26    and l.getStartLine() >= 39
27    and l.getEndLine() <= 150)
28
29    or (l.getFile().getBaseName() = "ServletPathMatches.java"
30    and l.getStartLine() >= 32
31    and l.getEndLine() <= 140))
32 select e,
33    e.toString() as element,
34    e.getAPrimaryQlClass() as elementType,
35    l.getFile().getBaseName() as file,
36    l.getStartLine() as startLine,
37    l.getEndLine() as endLine,
38    l.getStartColumn() as startColumn,
39    l.getEndColumn() as endColumn
```

