# OpenReview forum: "QLCoder: A Query Synthesizer For Static Analysis of Security Vulnerabilities"
_ICLR.cc/2026/Conference — ICLR 2026 Poster_

### Official Review · Reviewer_tkkv · 2025-10-25

**Soundness:** 3
**Presentation:** 3
**Contribution:** 2
**Rating:** 6
**Confidence:** 3

**Summary:**

The authors present FineNib – an agentic framework that automatically synthesizes queries in CodeQL, a powerful static analysis engine,
directly from a given CVE metadata. FineNib embeds an LLM in a synthesis loop with execution feedback, while constraining its reasoning using a custom MCP interface that allows structured interaction with a Language Server Protocol (for syntax guidance) and a RAG database (for semantic retrieval of queries and documentation). This approach allows FineNib to generate syntactically and  emantically valid security queries. The authors evaluate FineNib on 176 existing CVEs across
111 Java projects. Building upon the Claude Code agent framework, FineNib
synthesizes correct queries that detect the CVE in the vulnerable but not in the
patched versions for 53.4% of CVEs. In comparison, using only Claude Code
synthesizes 10% correct queries. The generated queries achieve an F1 score of
0.7. In comparison, the general query suites in IRIS (a recent LLM-assisted static
analyzer) and CodeQL only achieve F1 scores of 0.048 and 0.073, highlighting
the benefit of FineNib’s specialized synthesized queries

**Strengths:**

- The topic of CodeQL query generation for vulnerability detection is both timely and technically demanding. It presents significantly greater challenges than code generation for mainstream programming languages.
- The development of an MCP interface demonstrates substantial engineering effort. This contribution is valuable and enables structured reasoning and provides a foundation for future research in automated vulnerability analysis.
- FineNib achieves notably higher compilation, success, and F1 scores compared to existing methods.

**Weaknesses:**

- The authors claim that "Permitting free online search for vulnerability patterns or snippets similarly proved problematic." It would be helpful to provide empirical evidence or examples supporting this observation.
- While it is understandable that the presence of both vulnerable and fixed code versions simplifies the synthesis process, the realism of this assumption is uncertain. In practice, software updates often contain numerous unrelated edits (e.g., functional updates) to the affected CVE. How robust is the proposed method to such irrelevant changes between the two versions?
- The evaluation section refers to “CodeQL (version 2.22.2) query suites” as a baseline. This description is somewhat vague—could the authors clarify what specific query suites or rule sets were used? Since CodeQL itself does not autonomously generate queries, it would be useful to understand how these suites were selected and executed for comparison, and what they represent in the context of automated query synthesis.
- A broader weakness of the work is that it appears predominantly engineering/application-oriented, with limited discussion of principled methodologies for code generation in low-resource or domain-specific languages such as CodeQL. A stronger methodological foundation could enhance the generalizability and scientific contribution of the framework.

**Questions:**

Please address my concerns in **Weaknesses**.

---

> ### Author Response · Authors · 2025-11-20
>
> We thank the reviewer for their questions and have added more details related to the design decisions (Appendix B), FineNib’s vulnerability analysis (Section 3.3), and the CodeQL baseline evaluation (Appendix A). For any revisions mentioned, we have made the revision font blue in the revised, uploaded pdf.
>
> > **Q1: “The authors claim that ‘Permitting free online search for vulnerability patterns or snippets similarly proved problematic.’ It would be helpful to provide empirical evidence or examples supporting this observation.”**
>
> Free online search in our context appears to be error prone and simple mistakes easily lead to context rot and wasted token cost. We have added Appendix B to show detailed explanations and examples of problematic trajectories when the web search tools are available to the agent.
>
> > **Q2: While it is understandable that the presence of both vulnerable and fixed code versions simplifies the synthesis process, the realism of this assumption is uncertain. In practice software updates often contain numerous, unrelated edits to the affected CVE. How robust is the proposed method to such irrelevant changes between the two versions?**
>
> While it is true that software updates contain unrelated edits (test cases, configuration files, etc.), FineNib is robust to such changes. During FineNib’s pre-processing stage before the start of the agentic loop, vulnerability information along with patch information are stored into a vector database. As such, irrelevant edits are less likely to be retrieved during the agentic loop. We have updated Section 3.3 to provide more details.
>
> > **Q3: “The evaluation section refers to ‘CodeQL (version 2.22.2) query suites’ as a baseline. This description is somewhat vague–could the authors clarify what specific query suites or rule sets were used? Since CodeQL itself does not autonomously generate queries, it would be useful to understand how these suites were selected and executed for comparison, and what they represent in the context of automated query synthesis.”**
>
> For each CVE in our dataset, we use the corresponding official CodeQL CWE queries from the maintained repository (`java/ql/src/Security/CWE`). Each query is manually authored and continuously refined by CodeQL’s security analysis team and external contributors over multiple years, and represents the de-facto hand-written expert baseline for that CWE category.
>
> For evaluation, we map each CVE to its CWE classification(s) and run the associated CodeQL CWE query (or queries, if multiple apply) to check whether it can detect the vulnerability. We have added this clarifying description in Appendix A to the revised paper.
>
> > **Q4: "A broader weakness of the work is that it appears predominantly engineering/application-oriented, with limited discussion of principled methodologies for code generation in low-resource or domain-specific languages such as CodeQL. A stronger methodological foundation could enhance the generalizability and scientific contribution of the framework."**
>
> We agree that principled methodologies for generating code in low-resource DSLs are important. In our setting, we found that this challenge is fundamental: CodeQL has limited training data, a strict type system, and highly specialized idioms. We experimented with several methodology-driven approaches, including sampling-based prompting [1], iterative self-refinement without tools [2], and prompts augmented with CodeQL’s language specification. Across ~50 CVEs, these produced only 1 compilable query and 0 true positives.
>
> These failures highlight that the synthesis problem itself is nontrivial and that unassisted LLMs cannot reliably operate CodeQL. Our contribution is to formulate this problem and to demonstrate that a structured, tool-augmented approach (via MCP) provides a practical and generalizable methodology: FineNib produces compilable and specialized queries that detect the ground-truth vulnerability. We will add this discussion to the revised version of our paper.
>
> - [1] Brown, Bradley et al. “Large Language Monkeys: Scaling Inference Compute with Repeated Sampling.” ArXiv abs/2407.21787 (2024)
> - [2] Li, Penghui et al. “Automated Static Vulnerability Detection via a Holistic Neuro-symbolic Approach.” ArXiv abs/2504.16057 (2025)

---

> > ### Comment · Reviewer_tkkv · 2025-11-27
> >
> > Thank you for your rebuttal. After reading it carefully, I have the following concerns.
> >
> > > For evaluation, we map each CVE to its CWE classification(s) and run the associated CodeQL CWE query (or queries, if multiple apply) to check whether it can detect the vulnerability
> >
> > If I recall my experience with CodeQL correctly, the queries in the CodeQL repository are not intended to guarantee identification of all possible vulnerabilities under a given CWE. CWE categories are usually very coarse grained, and it is difficult for a single query or even a small set of queries to cover them comprehensively. My understanding is that the query suites are meant to serve as examples for authors who want to develop their own queries. Therefore, it is not scientifically sound to compare CWE focused queries (from the suite) with CVE focused queries (written by FineNib).

---

> > > ### Author Response · Authors · 2025-12-03
> > >
> > > We thank the reviewer for the valid point on comparing CodeQL CWE queries [1] with FineNib CVE queries. We argue that the comparison is fair. Related work such as MocQ [2] and IRIS [3] use CodeQL default queries as baseline comparisons. These default queries are used for industry-grade security analysis in continuous integration pipelines [4]. Note that for each CWE, there are multiple default CodeQL queries covering vulnerability variants (average 3 variant queries per CWE). In our dataset, 140/176 CVEs had corresponding CodeQL CWE default queries. CodeQL’s default query suite supports 22/38 CWEs represented by our dataset. Thus in our comprehensive baseline evaluation for each CVE with a CodeQL supported CWE, we ran each CWE’s query (or queries) and picked the best query’s results. As shown by the experiments, these queries expectedly have lower precision but they can still achieve positive recall on 33 out of 176 CVEs in our dataset. We will clarify this in the final version.
> > >
> > > - [1] CodeQL query suites, GitHub, https://docs.github.com/en/code-security/code-scanning/managing-your-code-scanning-configuration/codeql-query-suites
> > > - [2] Li, Penghui et al. “Automated Static Vulnerability Detection via a Holistic Neuro-symbolic Approach.” ArXiv abs/2504.16057 (2025)
> > > - [3] Li, Ziyang et al. “IRIS: LLM-Assisted Static Analysis for Detecting Security Vulnerabilities”, ICLR (2025)
> > > - [4] Configuring default setup for code scanning, GitHub https://docs.github.com/en/code-security/code-scanning/enabling-code-scanning/configuring-default-setup-for-code-scanning

---

### Official Review · Reviewer_5Esz · 2025-10-28

**Soundness:** 3
**Presentation:** 3
**Contribution:** 3
**Rating:** 6
**Confidence:** 4

**Summary:**

The paper presents FineNib, an agentic framework that synthesizes CodeQL queries directly from CVE text and patch signals. An LLM runs in a generate→execute→feedback loop and is constrained via a minimal MCP tool suite: (i) an LSP for syntax/API diagnostics and completion, and (ii) a RAG component to retrieve documentation, example queries, and small AST snippets. On CWE-Bench-Java (176 CVEs across 111 Java projects), the authors report a 53.4% success rate (fires on the vulnerable version and remains silent on the patched version) and strong F1, outperforming IRIS and off-the-shelf CodeQL query suites.

**Strengths:**

1) Translating CVE/patch information into executable static-analysis queries has clear value for regression testing, variant hunting, and patch verification in security engineering.
2) The constrained (MCP) tool set with LSP+RAG is a practical and novel combination that reduces hallucinated identifiers, version drift, and brittle syntax—common failure modes in code-query synthesis.
3) Treating “patched-version silence” as a first-class success criterion is well-motivated. The path-level metrics that intersect with patch deltas provide a principled notion of precision/recall beyond simple hit counts.

4) FineNib achieves substantial gains over IRIS and generic CodeQL suites on shared CVEs, and the reported overall success rate indicates meaningful practical headroom for automated coverage.

**Weaknesses:**

1) Limited external validity across languages and analysis engines; the study focuses on Java with CodeQL, leaving portability to Python/JS/C/C++ or to engines like Semgrep/Infer unclear.
2) Insufficient transparency on cost and throughput; the paper does not report detailed per-CVE runtime, tool-call/token breakdowns, or scaling with project size/CWE category.
3) Potential bias from metric design; scoring based on intersections with patch deltas may favor patch-aware synthesis and disadvantage baselines not modeling sanitizers or taint steps.

**Questions:**

1) Can you report a detailed cost profile per CVE (time cost, token usage), along with a breakdown by project size and CWE category?
2) What minimal changes are needed to port FineNib to Python/JS/C/C++? How would you replace or adapt LSP/RAG for other engines (e.g., Semgrep rules, Infer)?
3) Could you provide a controlled cross-agent evaluation where multiple LLMs use the same MCP interface and tool set, with end-to-end success rates and error categories?

---

> ### Author Response · Authors · 2025-11-20
>
> We thank the reviewer for their suggestions and have added RQ 4 (Section 4.6) on costs, discussion Section 5.1 for porting to other languages and SASTs, and more details on the cross-agent evaluation (Section 4.5) to the revised paper. For any revisions mentioned, we have made the revision font blue in the revised, uploaded pdf.
>
> > **Q1: “Can you report a detailed cost profile per CVE (time cost, token usage), along with a breakdown by project size and CWE category?”**
>
> We have added RQ 4 (Section 4.6) which discusses costs and runtime. We have also added a breakdown by project size and CWE category table in Appendix A (Table 7). We have also added how we derived the costs and runtime in Appendix A.
>
> > **Q2: What minimal changes are needed to port FineNib to Python/JS/C/C++? How would you replace or adapt LSP/RAG for other engines?**
>
> For Python/JS/C/C++ and all the other languages supported natively by CodeQL, we simply need to fetch the CodeQL documentation for the language and CodeQL QLpacks specific to the language to populate the vector database. We have added Section 5.1 in the revision for discussion on using FineNib for other languages and using FineNib with other SAST engines.
>
> > **Q3: “Could you provide a controlled cross-agent evaluation where multiple LLMs use the same MCP interface and tool set, with end to end success rates and error categories?”**
>
> We added more models for the cross-agent evaluation, and derived error categories for the cross-agent evaluation in RQ3 (Section 4.5). We have added success rates in Table 5 and detailed the error decompositions in Appendix A.

---

### Official Review · Reviewer_hinK · 2025-10-29

**Soundness:** 3
**Presentation:** 3
**Contribution:** 2
**Rating:** 2
**Confidence:** 4

**Summary:**

FineNib uses LLM's to synthesize CodeQL queries for a given CVE. Instead of doing it directly with an LLM, they integrate it into a loop using execution feedback to iteratively repair the CodeQL query that is generated. This execution feedback helps ensure that the query is correct syntactically and semantically. They show large improvements on CWE-Bench-Java compared to current state-of-the-art approaches, achieving an F1 score of .7.

**Strengths:**

I think there are some interesting ideas in this paper when it comes to neurosymbolic program analysis. I think that using execution feedback to guide the LLM to synthesize a query is a very good idea. There are many parts of program analysis that are tedious and difficult (writing CodeQL queries is a great example of this!). By incorporating an LLM into the loop and providing some feedback, I think we can make program analyses much more accessible and easier to write. I also liked the discussion section on alternative designs - it really helps highlight that more information is not always better when it comes to using these models. Also, performance is greatly improved using FineNib compared to other methods.

**Weaknesses:**

Unfortunately, where this paper falls apart for me is in the motivation. I don't understand why we need a query for one *specific* CVE, especially given the fact that that CVE has been patched. To my understanding, IRIS and other tools write queries that detect a general weakness pattern, like code injection (CWE-94). These queries can be used to detect that weakness pattern in other projects. But a CVE is just *one* instance of a CWE: "CVE-2024-12345: Buffer overflow in `foo()` of `libbar` 2.3.1 allows remote code execution".  I don't understand why we need a query for just that instance, *especially* given that it has been patched. It seems to me that the query would be useless from thereon out.

If the authors could show how FineNib could be used to discover *new* CVE's, that would be a huge improvement to this paper. But from the current version, it seems that FineNib is just detecting vulnerabilities that have already been detected and patched.

Some other small nitpicks:
- Figure 1 implies that FineNib needs only the CVE information. But it uses the CVE information, the vulnerable code, and the patched code.
- The first sentence of the introduction (line 030) states that security vulnerabilities are growing, but doesn't state how many CVE's were reported in 2023, 2022, etc.
- I'm a little confused at why IRIS has such a low F1 score. IRIS reports an F1 score of .177 in their paper, but it is .048 here. Specifically, the authors state that "The lack of true positive recall is why CodeQL and IRIS have significantly lower precision". Does that mean that IRIS still reports the vulnerability when it has been patched? I think this could be clearer.

**Questions:**

- Why do we need a CodeQL query for one specific vulnerability?
- Along the same lines, what can that query be used for besides the vulnerability that has been patched?
- Can the authors elaborate more on why IRIS has such a low F1 score? Is it because IRIS still reports the vulnerability in the patched version?

---

> ### Author Response · Authors · 2025-11-20
>
> We thank the reviewer for their feedback and have added more discussion on variant analysis (Section 5.2) to the revised paper. We also thank the reviewer for the “nitpicks” (they’re not! Thanks a bunch for pointing those out; we addressed them). For any revisions mentioned, we have made the revision font blue in the revised, uploaded pdf.
>
> **Nitpicks**:
> 1. We have added CVE numbers for 2022 and 2023 in the Introduction Section.
> 2. As for Figure 1, we have clarified in Section 3.2 that the vulnerable and patched versions are automatically derived from the CVE metadata. CVEs often will contain the commit hash of the patched version. The vulnerable version is the commit right before the patch.
>
> > **Q1: Why do we need a CodeQL query for one specific vulnerability? Why not a single query for a CWE?**
>
> General CWE-level queries inherently suffer from both low coverage and low precision. Prior work already documents this limitation: a single generic pattern frequently fails to capture the diverse manifestations, resulting in low recall, while the broadness of such patterns simultaneously leads to low precision due to over-approximating specifications (Table 2 of Li et al. [3]).
>
> This limitation is further exacerbated by the _heavy-tailed distribution_ of vulnerabilities: as shown by large-scale empirical studies (Iannone et. al. [1]), most vulnerability instances cluster, while the majority reside in a long tail of rare and highly heterogeneous patterns. Consequently, off-the-shelf static analyzers, including CodeQL, and IRIS [3], naturally optimize for these head classes, leaving substantial portions of the long tail under-analyzed.
>
> Our approach explicitly targets this gap. Rather than relying on a single general rule, we generate specialized, high-precision queries tailored to concrete variants, which (as demonstrated in our response to **Q2**) successfully support both variant analysis and regression testing, uncovering previously unknown vulnerabilities that general CWE queries miss. This is an approach also adopted by Yang et. al. [2].
>
> - [1] The Secret Life of Software Vulnerabilities: A Large-Scale Empirical Study, Iannone et. al., 2022
> - [2] KNighter: Transforming Static Analysis with LLM-Synthesized Checkers, Yang et. al., SOSP 2025
> - [3] IRIS: LLM-Assisted Static Analysis for Detecting Security Vulnerabilities, Li et. al., ICLR 2025
>
> > **Q2: “Along the same lines, what can the query be used for besides the vulnerability that has been patched?”**
>
> (**NEW EXPERIMENTS**) A CodeQL query for a specific vulnerability can be used for (1) regression testing, (2) variant analysis, and (3) patch validation (Figure 1). To further illustrate the point, since the start of the discussion period, we have found 2 previously unknown bugs in two different repositories with one FineNib generated CodeQL query from CVE-2025-52888. The bugs are reproducible with Proof-of-Concept exploits (PoCs), and we are currently in the process of reporting them to the developers. In addition, with the query presented in our motivating example (Figure 2), we were able to find 8 vulnerability variants in the same repository, all of which can be used for regression testing. We will update our revision as we find more during the discussion period. We added Section 5.2 to address the applications of the query.
>
> > **Q3: “Can the authors elaborate more on why IRIS has such a low F1 score?”**
>
> In our evaluation benchmark, we have not only the pre-2025 CVEs (111 CVEs from CWE-Bench-Java) but also the 65 new CVEs found in 2025 (Section 4.1.) We are able to reproduce the F1 score from the IRIS paper for the pre-2025 CVEs, but see a significant degradation in performance for the new ones. We note that the 65 new 2025-CVEs include more CWE types than the ones that IRIS currently supports. For a fair comparison, we added more CWE support to IRIS by using existing CodeQL queries as templates. However, CodeQL’s security queries don’t cover all of the CWEs in our benchmark, rendering both CodeQL and IRIS ineffective. As we discussed in Q1, these vulnerabilities exactly fall into the long-tail of the vulnerability pattern distribution, further illustrating the need for a FineNib-like tool to specialize vulnerability detection.

---

> > ### Comment · Reviewer_hinK · 2025-11-20
> >
> > I see! Those are solid use-cases. My only concern was that FineNib can only be used to detect vulnerabilities that have already been detected and nothing else, and that has been addressed very well. I think the "applications of finenib" section should be moved to the beginning though (maybe before illustrative example?). I think it's very helpful in grounding the rest of the paper and setting expectations for the reader that this tool isn't going to be used to discover brand-new CVE's.
> >
> > I'm raising my score to recommend acceptance. Thank you!

---

> > > ### Author Response · Authors · 2025-12-03
> > >
> > > Thanks for the suggestions and the recommendation of the paper. We will improve our organization of the sections in the revised version. To clarify, FineNib discovers brand new bugs as we discussed in our rebuttal.

---

### Official Review · Reviewer_n8Qx · 2025-11-01

**Soundness:** 3
**Presentation:** 4
**Contribution:** 3
**Rating:** 8
**Confidence:** 4

**Summary:**

The paper introduces FineNib, an agentic framework that generates CodeQL queries from CVE metadata. To mitigate the pitfalls of pure LLM-based approaches, FineNib utilizes iterative reasoning with syntax guidance and a RAG database for semantic retrieval of queries and documentation. FineNib is evaluated on CWE-Bench-Java with Claude Sonnet 4, where 100% of generated queries complied, and 53.4% successfully detected the vulnerability. This is a much-improved result over the vanilla agentic baselines.

**Strengths:**

+ The application of script generation for vulnerability detection is a novel application, and with the help of RAC, the approach can mitigate the issues with low-resource static analysis query languages.
+ The approach shows great efficiency, outperforming vanilla LLMs and original CodeQL analysis by a big margin.

**Weaknesses:**

- The evaluation shows that FineNib can successfully generate good queries. However, the evaluation of a single LLM makes it somewhat incomplete. Claude Sonnet 4 is a very powerful, expensive, closed-source LLM; knowing the effectiveness of FineNib with a weaker open-source LLM would make the evaluation more complete.
- Since agentic approaches tend to be more costly due to their iterative refinement process, a cost vs effectiveness analysis could inform the reader of the cost per performance gain. Having an additional open-source LLM in the evaluation with this cost analysis can also paint a more complete picture of the overall efficiency of FineNib.
- Since the baseline agents with Claude Sonnet 4 are not presented in Table 4, it is hard to follow (it is kind of presented in the ablation study). Having Claude Sonnet 4 as the baseline agent would help highlight the effectiveness of the contribution more.

**Questions:**

1. It is not clear from the text if the performance was measured under a regression testing scenario or a variant analysis. It is important to know the performance separately between these two scenarios. What is the performance of FineNib in the variance analysis scenario (i.e., generate queries from one issue and detect other issues of the same type)?
2. Since the static analysis query is a low-resource language, would finetuning a smaller model specifically to generate the code for this language be a feasible approach? Would this potentially reduce the operational cost for a trade-off of a slightly larger cost upfront?
3. Could an additional open-source LLM be added for the evaluation to create a more complete evaluation picture?
4. Do you have some cost statistics that you can quickly perform a cost vs effectiveness analysis of FineNib?

---

> ### Author Response · Authors · 2025-11-20
>
> We thank the reviewer for their clarifying questions and suggestions. We have added RQ 4 (Section 4.6) to address costs and runtime, and an open source model in our evaluation (Section 4.5)  in the revised uploaded pdf. We also updated ablation Table 4 to better show Claude Code without FineNib compared to FineNib’s other ablations. Finally, we added Section 5.2 on the applications of FineNib which discuss regression testing and variant analysis. For any revisions mentioned, we have made the revision font blue in the revised, uploaded pdf.
>
> > **Q1: It is not clear from the text if the performance was measured under a regression testing scenario or a variant analysis. It is important to know the performance separately between these two scenarios. What is the performance of FineNib in the variance analysis scenario?**
>
> We have added Section 5.2, which discusses using FineNib for regression testing and variant analysis. In short, since the beginning of the discussion period, we have found 2 previously unknown CWE-611 bugs with variant analysis and are currently in the process of reporting them. We will update these numbers as we find more bugs during the discussion period. Further illustrating the point, with the query presented in our motivating example (Figure 2), we were able to find 8 vulnerability variants in the same repository.
>
> > **Q2: “Since the static analysis query is a low-resource language, would fine-tuning a smaller model specifically to generate the code for this language be a feasible approach? Would this potentially reduce the operational cost for a trade-off of a slightly large cost upfront?”**
>
> While fine-tuning a smaller model for CodeQL is a feasible path to reduce operational cost, we envision the following challenges: (1) CodeQL is still a developing tool with frequent breaking changes in syntax and language-specific constructs, and (2) CodeQL programs are fragmented across different programming languages, query types, and abstraction levels. Solving such challenges during fine-tuning would be a promising future direction of this paper. In fact, FineNib could be used to generate more QL queries to fine-tune a model.
>
> > **Q3: “Could an additional open-source LLM be added for the evaluation to create a more complete evaluation picture?”**
>
> (**NEW EXPERIMENTS**) Yes. We have added an open source baseline model to our evaluation (see Table 5 and Section 4.5). We used `gpt-oss:20b` with the Codex CLI. While we attempted DeepSeek-R1, Qwen3-Coder:30b, and gpt-oss:20b, we find that the former two either does not provide tooling support or is ineffective in using the LSP or writing QL files.
>
> > **Q4: “Do you have some cost statistics that you can quickly perform a cost vs. effectiveness analysis of FineNib?”**
>
> Yes we have added another RQ in Section 4.6, which discusses cost statistics and have added a more detailed table in Appendix A - see Table 7. We have also added how we derived the costs and runtime in Appendix A.

---

### Official Review · Reviewer_XjUp · 2025-11-01

**Soundness:** 3
**Presentation:** 4
**Contribution:** 3
**Rating:** 6
**Confidence:** 3

**Summary:**

This paper presents FineNib, a system that automatically synthesizes CodeQL vulnerability queries from CVE text and vulnerable/patched code. It uses an LLM agent with execution feedback and structured tool access (CodeQL LSP + RAG) to iteratively refine queries. On 176 Java CVEs, FineNib achieves a high rate of successful queries and good F1, substantially outperforming IRIS and built-in CodeQL rules.

**Strengths:**

1. Tackles a practical and timely challenge in security automation.
2. Clear design: LSP for syntax, RAG for semantics, feedback loop for correctness.
3. Strong results across real-world projects; large margin over baselines.
4. Good ablations showing importance of LSP + retrieval.

**Weaknesses:**

1. Only Java CVEs are supported; unclear how easily this extends to other CodeQL languages (C/C++, JS, Python). Authors mention RAG replacement, but actual cross-language results are not shown.
2. The system leans on Claude’s coding agent + CodeQL LSP. It is unclear how reproducible this is without Anthropic tooling or whether open-model support is viable (Gemini/GPT baselines fail).
3. Iterative refinement with code execution can be slow. Wall-clock time, computational budget, and iteration cost are not reported.
4. The metric assumes "raises alarm on vulnerable version and silent on patched version" as correctness. But some vulnerabilities have multiple exploitation paths; authors explicitly allow false positives, which may inflate success numbers.
5. The work is strong for static analysis automation, but the conceptual novelty in LLM reasoning or program analysis theory is incremental. It feels like an engineering system rather than theoretical innovation.

**Questions:**

1. What is the average runtime per CVE?
2. How sensitive is the system to CodeQL version changes?
3. Could the same iterative loop work with a weaker LLM if aided by stronger program-analysis constraints?
4. For multi-file or multi-snapshot vulnerabilities, how do you avoid overfitting to the specific patch change?
5. Can the pipeline detect variants of vulnerabilities (not only the exact CVE instance)?

---

> ### Author Response · Authors · 2025-11-20
>
> We thank the reviewer for their suggestions and have added RQ 4 (Section 4.6) on costs & runtime, a discussion on portability of FineNib (Section 5.1), and further clarification of the vulnerability analysis phase (Section 3.3) to the revised paper. The questions regarding runtime and portability are insightful and we agree should be addressed. We have added a weaker and open-source model `gpt-oss:20b` to our agent baselines. For any revisions mentioned, we have made the revision font blue in the revised, uploaded pdf.
>
> > **Q1: “How sensitive is the system to CodeQL version changes?”**
>
> FineNib is robust to CodeQL version changes. We have added Section 5.1 to discuss FineNib’s versatility regarding switching CodeQL versions. In our own experience, we were able to reproduce comparable performance results with FineNib even after a major version update of CodeQL (2.22.*->2.23.*).
>
> Specifically, when changing CodeQL versions, the system has to obtain CodeQL documentation and queries related to the specified version. We provide a python script that automatically fetches all the necessary resources to populate the vector database. In short, the procedure for version upgrade is fully automated.
>
> > **Q2: “Could the same iterative loop work with a weaker LLM if aided by stronger program-analysis constraints?”**
>
> (**NEW EXPERIMENTS**) Please see Table 5 in the revised document for our new experiment on an open-source model baseline (`gpt-oss:20b`). To make sure that a weaker LLM can still function well, we break down the existing prompts into smaller, more constrained tasks. At the end, synthesized subqueries for sources, sinks, taint steps, and sanitizers are composed together for the final CodeQL query.
>
> As for stronger program-analysis constraints: yes, finer-grained semantic checks on each component of a query would likely provide even stronger feedback for weaker LLMs. However, performing program analysis on CodeQL queries themselves is a separate and nontrivial research problem. We view this as promising future work that could improve both weak and strong LLMs, but it is orthogonal to the contribution of the current paper.
>
> > **Q3: “Can the pipeline detect variants of vulnerabilities (not only the exact CVE instance)?”**
>
> Yes. Since the start of the discussion period, we were able to detect 2 previously unknown CWE-611 bugs (which are both variants of CVE-2025-52888) with one FineNib synthesized query. We have successfully reproduced the bugs with Proof-of-Concepts (PoCs) and are currently in the process of reporting them to the developers. Additionally, using the query presented in our motivating example (Figure 2), we were able to find 8 variants in the same repository. We will update these numbers as we find more vulnerabilities during the discussion period. We have also added Section 5.2 to discuss using FineNib for variant analysis and other applications.
>
> > **Q4: “For multi-file or multi-snapshot vulnerabilities, how do you avoid overfitting to the specific patch change?”**
>
> In general, the design of FineNib encourages abstraction which would reduce overfitting. Patch changes can be inherently multi-file and multi-snapshot and typically patch changes are captured only in the sanitizer part of the query. The source and sink specifications do not necessarily need to correspond to patch changes, thus avoiding the overfitting. Our updated Section 3.3 elaborates on this topic, while our variant analysis (Q3) further demonstrates the generalizability of our generated queries.
>
> > **Q5: “What is the average runtime per CVE?”**
>
> The average runtime per CVE is 3712 seconds, though we find that successful synthesis campaigns demand much less time than unsuccessful ones. We have added RQ4 (Section 4.6) for discussing runtime and costs, and have added a more detailed table on costs and runtime in Appendix A (Table 7). We have also added how we derived the costs and runtime in Appendix A.

---

### Author Response · Authors · 2025-12-03

We thank the reviewers and the ACs for the discussion period. We summarize the feedback discussed and revisions made in the uploaded PDF. Note that in the uploaded PDF, all revisions are in blue font. We are happy to provide any further clarification if needed.

The strengths acknowledged by the reviewers include:
- Automating the time-consuming challenge of writing CodeQL queries. (XjUp, hinK, tkkv)
- FineNib significantly outperforms baselines on real-world projects. (XjUp, n8Qx, 5Esz, tkkv)
- FineNib is robust to issues with LLMs generating low resource languages. (n8Qx, 5Esz)
- Exposing the CodeQL language server and a RAG database using the Model Context Protocol tool interfaces is a novel and practical contribution. (n8Qx, 5Esz, tkkv)

During the discussion period we addressed the following concerns:
- We demonstrated the applications of FineNib queries such as variant analysis and regression testing by adding Section 5.2. Within the first week of the discussion period we found 2 unknown bugs using one FineNib query, as a response to the reviewers asking if FineNib queries could be used for variant analysis. (XjUp, n8Qx, hinK)
- We added more experiments for our cross agent baselines in Section 4.5 - including an open source model. We also provided more metrics to the cross agent experiments and analyzed common error categories. (XjUp, n8Qx, 5Esz)
- We provided runtime and costs of using FineNib by adding RQ 4 (Section 4.6). (XjUp, n8Qx, 5Esz)
- We addressed the portability of FineNib to other versions of CodeQL and for analyzing languages other than Java by adding Section 5.1. (XjUp, 5Esz)
- We clarified how we use patch diffs during the vulnerability analysis phase in Section 3.3. (XjUp, tkkv)
- We added Appendix B to further explain design decisions such as caching CVE data instead of allowing an agent to use free web search. (tkkv)

---

### Meta-Review · Area_Chair_kzig · 2025-12-30

**Summary:**

The reviewers mentioned the following strengths:
- Relevance of the problem the authors focused on.
- Clear design.
- Good results.
- Several good ideas.

The reviewers pointed out the following weaknesses:
- Only Java. Unclear how to extend CodeQL to other languages. No cross language results shown. **Partially Addressed**. Evaluation no evaluation on other languages.
- Unclear how to reproduce the results without Anthropic tool calling or whether open models succeed. **Partially Addressed**
- Wall-clock time, computational budget and iteration cost are not reported. **Addressed**
- Some vulnerabilities have multiple exploitation paths; authors explicitly allow false positives (may inflate success numbers). Unclear how to avoid overfitting. **Partially Addressed.** The shortcomings here appear to be systemic.
- The conceptual novelty is incremental. More of an engineering system rather then theoretical innovation. **Partially Addressed**
- Unclear motivation. Why do we need a query for one specific CVE? **Addressed**
- Unclear why IRIS has such a low F1 score. **Addressed**
- Only single LLM evaluated. **Addressed**
- A cost vs effectiveness analysis would be informative. **Addressed**
- Potential bias from metric design; scoring based on intersections with patch deltas may favor patch-aware synthesis and disadvantage baselines not modeling sanitizers or taint steps. **Not Addressed**

The reviewers asked the following questions:
- Unclear how sensitive the system is to CodeQL version changes. **Addressed**
- Could the same iterative loop work with a weaker LLM if aided by stronger program-analysis constraints? **Partially addressed**. Some experiments (just GPT-OSS 20B) where provided, though more would have been beneficial.
- Unclear if regression testing or variant analysis was the scenario in the evaluation. **Addressed**
- Would fine-tuning a small model to generate CodeQL be viable approach?  **Addressed**
- While it is understandable that the presence of both vulnerable and fixed code versions simplifies the synthesis process, the realism of this assumption is uncertain. In practice, software updates often contain numerous unrelated edits (e.g., functional updates) to the affected CVE. How robust is the proposed method to such irrelevant changes between the two versions? **Addressed**
- The evaluation section refers to “CodeQL (version 2.22.2) query suites” as a baseline. This description is somewhat vague—could the authors clarify what specific query suites or rule sets were used? Since CodeQL itself does not autonomously generate queries, it would be useful to understand how these suites were selected and executed for comparison, and what they represent in the context of automated query synthesis. **Addressed**

**Reviewer Concerns:**

Most reviewer concerns where addressed as indicated above.

**Reviewer Scores:**

- XjUp: retain -- 6
- n8Qx: retain -- 8
- hinK: raise -- 6
- 5Esz: retain -- 6

---

### Decision · Program_Chairs · 2026-01-26

Accept (Poster)